# Generation of contractile actomyosin bundles depends on mechanosensitive actin filament assembly and disassembly

**Sari Tojkander[1], Gergana Gateva[1], Amjad Husain[2], Ramaswamy Krishnan[2], Pekka Lappalainen[1]\***

[1]Institute of Biotechnology, University of Helsinki, Helsinki, Finland; [2]Beth Israel Deaconess Medical Center, Harvard Medical School, Boston, United States

**Abstract** Adhesion and morphogenesis of many non-muscle cells are guided by contractile actomyosin bundles called ventral stress fibers. While it is well established that stress fibers are mechanosensitive structures, physical mechanisms by which they assemble, align, and mature have remained elusive. Here we show that arcs, which serve as precursors for ventral stress fibers, undergo lateral fusion during their centripetal flow to form thick actomyosin bundles that apply tension to focal adhesions at their ends. Importantly, this myosin II-derived force inhibits vectorial actin polymerization at focal adhesions through AMPK-mediated phosphorylation of VASP, and thereby halts stress fiber elongation and ensures their proper contractility. Stress fiber maturation additionally requires ADF/cofilin-mediated disassembly of non-contractile stress fibers, whereas contractile fibers are protected from severing. Taken together, these data reveal that myosin-derived tension precisely controls both actin filament assembly and disassembly to ensure generation and proper alignment of contractile stress fibers in migrating cells.

**\*For correspondence:** pekka. lappalainen@helsinki.fi

**Competing interests:** The authors declare that no competing interests exist.

## Introduction

Cell migration is essential for embryonic development, wound healing, immunological processes and cancer metastasis. Cell migration is driven by assembly and disassembly of protrusive and contractile actin filament structures. The force in protrusive actin filament structures, including lamellipodium and filopodia at the leading edge of cell, is generated through actin polymerization against the plasma membrane. In contractile actin filament bundles, such as stress fibers, the force is generated by sliding of bipolar myosin II bundles along actin filaments. Notably, whereas the assembly-mechanisms of protrusive actin filament structures are relatively well understood, general principles underlying the assembly of contractile actomyosin bundles have remained elusive (*Pollard and Cooper, 2009*; *Bugyi and Carlier, 2010*; *Michelot and Drubin, 2011*; *Burridge and Wittchen, 2013*).

The most prominent contractile actomyosin structures in most cultured non-muscle cells are stress fibers. Beyond cell migration, stress fibers guide adhesion, mechanotransduction, endothelial barrier integrity, myofibril assembly, and receptor clustering in T-lymphocytes (*Burridge and Wittchen, 2013*; *Wong et al., 1983*; *Sanger et al., 2005*; *Tojkander et al., 2012*; *Yi et al., 2012*). Due to their intrinsic properties, stress fibers have become an important model system for studying the general principles by which contractile actomyosin bundles are assembled in cells. Stress fibers can be divided into three main categories based on their protein compositions and interactions with focal adhesions (*Small et al., 1998*). *Dorsal (radial) stress fibers* are connected to focal adhesions at their distal ends and rise towards the dorsal surface of the cell at their proximal region (*Hotulainen and Lappalainen, 2006*). They elongate through vectorial actin polymerization at focal adhesions (i.e. coordinated polymerization of actin filaments, whose rapidly elongating barbed ends are facing the

**eLife digest** Muscle cells are the best-known example of a cell in the human body that can contract. These cells contain bundles of filaments made of proteins called actin and myosin, which can generate pulling forces. However, many other cells in the human body also rely on similar "contractile actomyosin bundles" to help them stick to each other, to maintain the correct shape or to migrate from one location to another. These bundles in the non-muscle cells are often called "ventral stress fibers".

Ventral stress fibers develop from structures commonly referred to as "arcs". Previous work has clearly established that ventral stress fibers are sensitive to mechanical forces. However, the underlying mechanism behind this process was not known, and it remained unclear how external forces could promote these actomyosin bundles to assemble, align and mature.

Tojkander et al. documented the formation of ventral stress fibers in migrating human cells grown in the laboratory. This revealed that pre-existing arcs fuse with each other to form thicker and more contractile actomyosin bundles. The formation of these bundles then pulls on the two ends of the stress fibers that are attached to sites on the edges of the cell.

Tojkander et al. also showed that this tension inactivates a protein called VASP, which is also found at these sites. Inactivating VASP inhibits the construction of actin filaments, which in turn stops the stress fibers from elongating and allows them to contract. Further experiments then revealed that ventral stress fibers are maintained and can even become thicker under a sustained pulling force. Conversely, stress fibers that were not under tension were decorated by proteins that promote the disassembly of actin filaments. This subsequently led to the disappearance of these fibers.

Future studies could now examine whether the newly identified pathway, which allows mechanical forces to control the assembly and alignment of stress fibers, is conserved in other cell-types. Furthermore, and because the assembly of such mechanosensitive actomyosin bundles is often defective in cancer cells, it will also be important to study this pathway's significance in the context of cancer progression.

focal adhesion, is responsible for growth of dorsal stress fibers). These actin filament bundles do not contain myosin II, and dorsal stress fibers are thus unable to contract (*Hotulainen and Lappalainen, 2006*; *Cramer et al., 1997*; *Tojkander et al., 2011*; *Oakes et al., 2012*; *Tee et al., 2015*). However, dorsal stress fibers interact with contractile *transverse arcs* and link them to focal adhesions. Transverse arcs are curved actin bundles, which display periodic α-actinin – myosin II pattern and undergo retrograde flow towards the cell center in migrating cells. They are derived from α-actinin- and tropomyosin/myosin II- decorated actin filament populations nucleated at the lamellipodium of motile cells (*Hotulainen and Lappalainen, 2006*; *Tojkander et al., 2011*; *Burnette et al., 2011*; *2014*). In fibroblasts and melanoma cells, filopodial actin bundles can be recycled for formation of transverse arc –like contractile actomyosin bundles (*Nemethova et al., 2008*; *Anderson et al., 2008*). *Ventral stress fibers* are defined as contractile actomyosin bundles, which are anchored to focal adhesions at their both ends. Despite their nomenclature, the central regions of ventral stress fibers can bend towards the dorsal surface of the lamellum (*Hotulainen and Lappalainen, 2006*; *Schulze et al., 2014*). Migrating cells display thick ventral stress fibers that are typically oriented perpendicularly to the direction of migration, and thinner ventral stress fibers that are often located at the cell rear or below the nucleus. At least the thick ventral stress fibers, which constitute the major force-generating actomyosin bundles in migrating cells, are derived from the pre-existing network of dorsal stress fibers and transverse arcs. However, the underlying mechanism has remained poorly understood (*Burridge et al., 2013*; *Hotulainen and Lappalainen, 2006*).

Stress fibers and focal adhesions are mechanosensitive structures. Stress fibers are typically present only in cells grown on rigid substrata and they disassemble upon cell detachment from the matrix (*Mochitate et al., 1991*; *Discher et al., 2005*). Furthermore, after applying fluid shear stress, stress fibers align along the orientation of flow direction in endothelial cells (*Sato and Ohashi, 2005*). Also focal adhesions develop only on rigid surfaces, and applying external tensile force promotes their enlargement (*Chrzanowska-Wodnicka and Burridge, 1996*; *Pelham et al., 1999*;

*Riveline et al., 2001*). Focal adhesions contain several mechano-sensitive proteins, including talin, filamin and p130Cas, whose activities and interactions with other focal adhesion components can be modulated by forces of ~~10–50 pN range (*Sawada et al., 2006*; *del Rio et al., 2009*; *Ehrlicher et al., 2011*). Furthermore, the protein compositions of focal adhesions are regulated by tension supplied by myosin II activity and external forces applied to the cell (*Zaidel-Bar et al., 2007*; *Kuo et al., 2011*; *Schiller et al., 2011*). Importantly, despite wealth of information concerning mechanosensitive focal adhesion proteins, possible effects of tensile forces on actin filament assembly at focal adhesions have remained elusive. Furthermore, the mechanisms by which tension contributes to the alignment of stress fibers and actin dynamics within these actomyosin bundles have not been reported.

Here we reveal that formation of mature contractile actin bundles from their precursors is a mechanosensitive process. We show that arc fusion during centripetal flow is accompanied by increased contractility that inhibits vectorial actin polymerization at focal adhesions through AMPK-mediated phosphorylation of VASP, thus insuring formation of ventral stress fibers. Conversely, activation of AMPK allows generation of contractile ventral stress fibers in cells growing on compliant matrix, where their formation is normally prevented. Furthermore, we provide evidence of mechanosensitive actin filament disassembly by ADF/cofilins during stress fiber assembly. These data provide support to a new mechanobiological model explaining the principles of assembly and alignment of ventral stress fibers in migrating cells.

## Results

### Transverse arcs fuse with each other during centripetal flow

Transverse arcs are generated from actin filament arrays at the lamellipodium —— lamella interface (*Tojkander et al., 2011*; *Shemesh et al., 2009*; *Burnette et al., 2011*). During their assembly, thin arcs associate with elongating dorsal stress fibers to form a spider-net -like structure (*Figure 1—figure supplement 1A and 1B*; *Tojkander et al., 2011*). This network, consisting of several non-contractile dorsal stress fibers and multiple thin arcs, flows towards the cell center and matures to thick, contractile ventral stress fibers through a mechanism that has remained poorly understood (*Hotulainen and Lappalainen, 2006*). Interestingly, proper stress fiber network does not form in cells grown on compliant matrix (*Discher et al., 2005*; *Prager-Khoutorsky et al., 2011*), but whether the assembly of all above-mentioned stress fiber categories, or only a specific one, is mechanosensitive has not been reported. By plating U2OS cells on soft (0.5 kPa) and stiff (64 kPa) substrata, we revealed that dorsal stress fibers and arcs are also present in cells grown on compliant matrix. In contrast, ventral stress fiber assembly is compromised under these conditions (*Figure 1A*). While 89% of cells plated on 64 kPa matrix contained ventral stress fibers, only 10% of cells plated on 0.5 kPa matrix exhibited ventral stress fibers as defined by presence of straight, contractile actin bundles connected to focal adhesions at each end. Thus, generation of ventral stress fibers appears to be the mechanosensitive phase in the formation of the stress fiber network.

To reveal how ventral stress fibers are derived from arcs and to elucidate the mechanosensitive basis of this process, we examined the dynamics of the stress fiber network in U2OS cells, where all three stress fiber categories can be readily visualized by live-cell microscopy (*Hotulainen and Lappalainen, 2006*). We first followed this process by using GFP-calponin-3 (CaP3), which compared to other stress fiber components allows better visualization of thin arc precursors. Interestingly, live-imaging of GFP-CaP3 -transfected cells revealed that the thin arc precursors fused with each other to form thicker actomyosin bundles during their flow towards the cell center (*Figure 1B*; *Figure 1—figure supplement 1B*). Fusion appeared to often initiate at the sites where arcs were connected to elongating dorsal stress fibers (*Figure 1C*). Live-imaging of cells expressing CFP-α-actinin and YFP-tropomyosin-4 demonstrated that homotypic coalescence of tropomyosin-4/myosin II foci and α-actinin foci of adjacent arcs occurred during the fusion process in all observed cases (*Figure 1D*). Thus, thin arc precursors fuse with each other during centripetal flow to generate thicker actomyosin bundles, where the periodic α-actinin — myosin II pattern is retained.

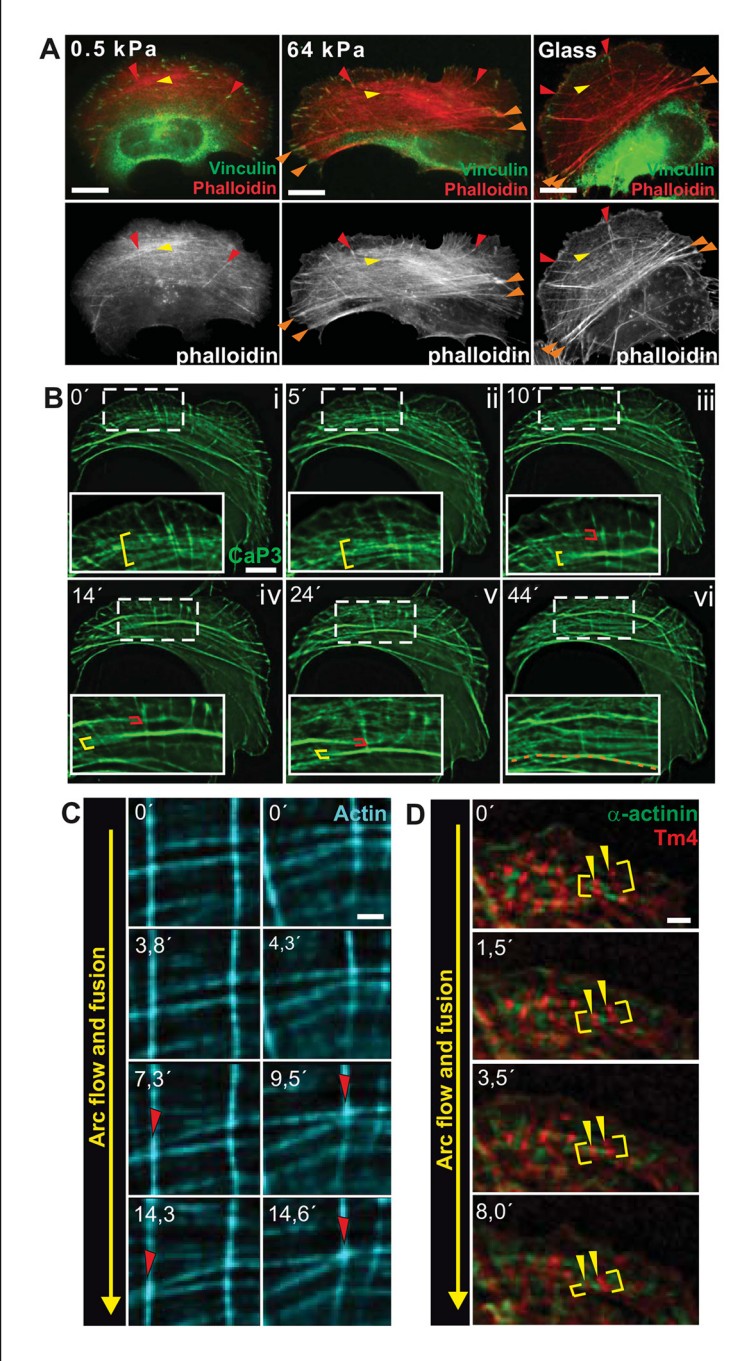

**Figure 1.** Transverse arcs fuse during centripetal flow to generate a contractile ventral stress fiber. (A) U2OS cells have three subtypes of stress fibers: Dorsal stress fibers (red arrowheads), which are attached to focal adhesion at their distal end; Transverse arcs (yellow arrowheads), which are curved actomyosin bundles oriented parallel to the leading edge; Ventral stress fibers (orange arrowheads), which are thick contractile bundles connected to focal adhesions at both ends. All three stress fiber categories are present in cells grown on glass or on stiff ($E$ = 64 kPa) silicone matrix, whereas assembly of contractile ventral stress fibers is compromised in cells grown on soft (0.5 kPa) matrix. (B) Live-imaging of U2OS cells expressing GFP-calponin-3 (CaP3) revealed that transverse arcs fuse with each other during centripetal flow to form thicker actomyosin bundles. Red and yellow brackets highlight fusing arcs, and the orange dashed line indicates the thick ventral stress fiber derived from the fusing arcs. (C) Arc fusion often initiates at the connection points of dorsal stress fibers and transverse arcs (indicated by red arrowheads). Two separate video frame series are shown in the panels. In the images, dorsal stress fibers and arcs are oriented vertically and horizontally, respectively. Stress fibers were visualized by expression of GFP-actin. Bar, 1 µm. (D) Live

*Figure 1 continued on next page*

*Figure 1 continued*

imaging of YFP-Tm4 and CFP-α-actinin-1 expressing U2OS cell reveals that homotypic coalescence of adjacent Tm4 and α-actinin foci occurs during arc fusion, thus allowing to retain the periodic pattern of transverse arcs. Yellow brackets indicate fusing arcs and yellow arrowheads highlight pairs of fusing α-actinin-1 foci. Bar, 1 μm.

The following figure supplement is available for figure 1:

**Figure supplement 1.** Fusion of transverse arcs.

## Transverse arc fusion is accompanied by increased contractility of actomyosin bundles and alignment of distal focal adhesions

Traction force microscopy was applied to examine whether arc fusion during centripetal flow is accompanied by changes in their contractility. These experiments revealed that thick ventral stress fibers exhibit stronger traction forces to focal adhesions as compared to forces applied by dorsal stress fibers (*Figure 2A and B*), similarly to what was recently demonstrated with model-based traction force microscopy by *Soine et al. (2015)*. Furthermore, spacing between individual CaP-3 foci, which co-localize with α-actinin in stress fibers (*Small and Gimona, 1998*), decreased as the arcs flowed towards the cell center and become thicker as detected both from several fixed samples and live-cell imaging experiments (representative examples are shown *Figure 1—figure supplement 1C and D*). This correlates well with the increased contractility of the structures (*Aratyn-Schaus et al., 2011*).

Transverse arcs are typically connected to several focal adhesion-attached dorsal stress fibers along their length (*Hotulainen and Lappalainen, 2006*). To elucidate how increased contractility of arcs affects the associated focal adhesions, we examined possible changes in adhesion alignment during the arc maturation process. These experiments revealed that the 'distal' focal adhesions, linked via dorsal stress fibers to the ends of the arc, turned and aligned along the direction of arc. In contrast, focal adhesions linked to the central region of the arc did not display similar alignment during the process. Alignment of 'distal' focal adhesions correlated with arc fusion, and was accompanied by enlargement of adhesions (*Figure 2—figure supplement 1A and B*). Thus, arc fusion during centripetal flow correlates with their increased contractility, consequent enlargement of distal focal adhesions and their alignment along the direction of the actomyosin bundle. Eventually, this leads to formation of a directed ventral stress fiber, containing one properly aligned large focal adhesion at its both ends.

## Tension provided by myosin II inhibits vectorial actin polymerization at focal adhesions

Dorsal stress fibers elongate through actin polymerization at focal adhesions. In U2OS cells, this 'vectorial' actin polymerization promotes elongation of the actin filament bundle with a rate of ~0.25 μm/min (*Hotulainen and Lappalainen, 2006*). In addition, focal adhesions may contain other actin filament populations that are not directly associated with vectorial actin polymerization and consequent elongation of dorsal stress fibers. This is because several tropomyosin isoforms, which are likely to decorate distinct actin filament populations, localize to focal adhesions (*Tojkander et al., 2011*) and because several proteins involved in actin polymerization regulate actin dynamics at focal adhesions (e.g. *Hotulainen and Lappalainen, 2006*; *Skau et al., 2015*). Furthermore, FRAP experiments performed at focal adhesions show rapid, uniform recovery of GFP-actin fluorescence (*Videos 1* and *2*; *Figure 2—figure supplement 2*), whereas FRAP experiments performed at dorsal stress fiber regions below focal adhesions exhibit treadmilling-like recovery that is indicative of vectorial actin polymerization (*Hotulainen and Lappalainen, 2006*; *Tee et al., 2015*).

Because contractility promotes focal adhesion enlargement and alignment during maturation of arcs to ventral stress fibers, we examined whether this process would be accompanied by alterations in vectorial actin polymerization at focal adhesions. Fluorescence-recovery-after-photobleaching (FRAP) was first applied to visualize the recovery of GFP-actin signal within actin filament bundles of dorsal and ventral stress fibers. Region of interest was chosen beneath focal adhesions to exclude other focal adhesion associated actin filament populations that are not directly involved in vectorial actin polymerization and elongation of stress fibers. As previously reported, elongation of a bright

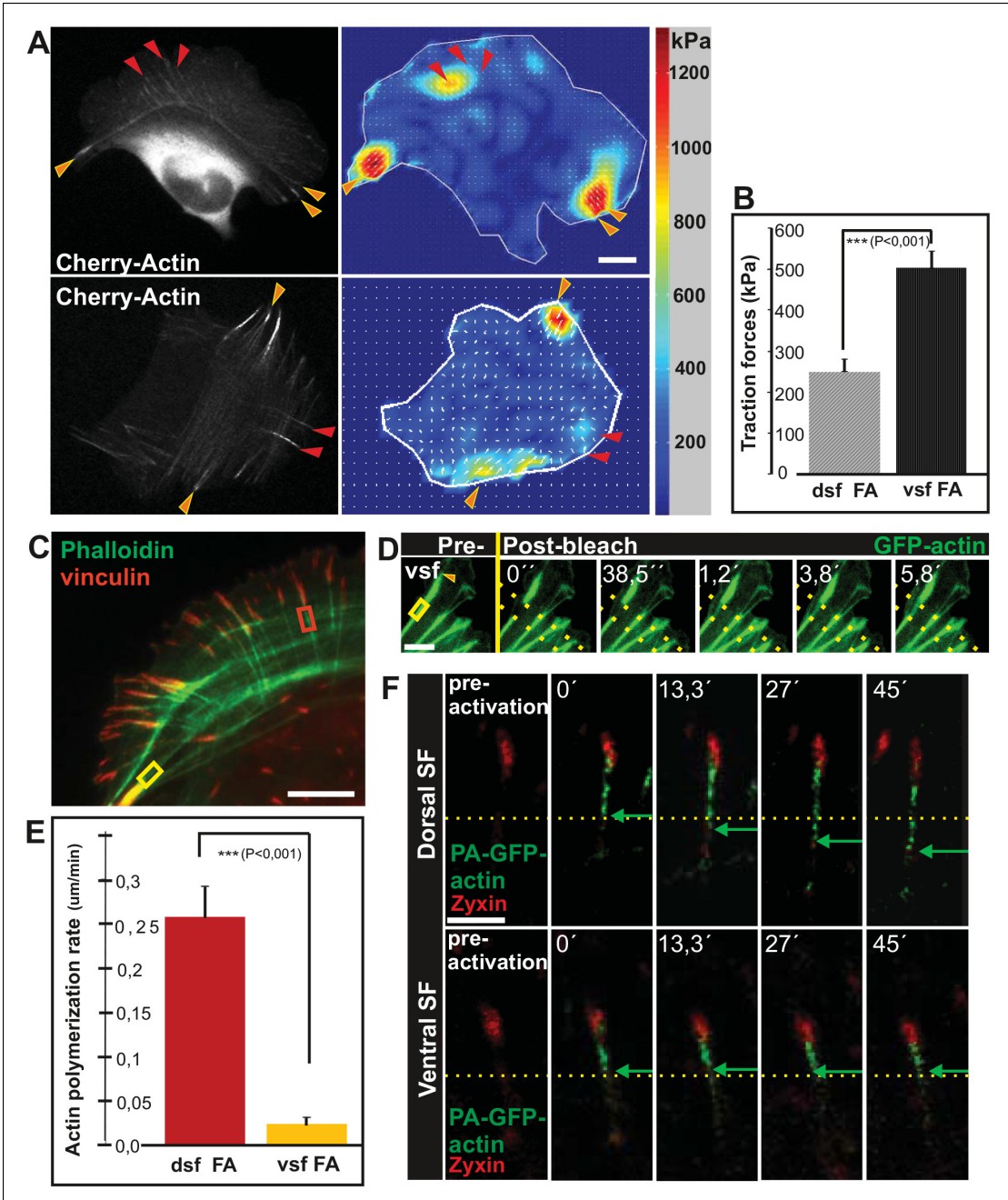

**Figure 2.** Vectorial actin polymerization at focal adhesions halts upon increased contractility and formation of ventral stress fibers. (**A**) Representative images of U2OS cells grown on 26 kPa polyacrylamide dishes with fluorescent nanobeads together with the corresponding force maps. Adhesions located at the ends of ventral stress fibers (orange arrowheads) apply stronger forces to their substrate compared to adhesions located at the ends of dorsal stress fibers (red arrowheads). Bar, 10 um. (**B**) Quantification of traction forces at adhesions located in the ends of dorsal and ventral stress fiber adhesions. Mean +/- SEM, n = 20 cells, 4–8 adhesions per cell. (**C**) Recovery of GFP-actin signal was measured next to dorsal stress fiber adhesions (dsf FA, red box) and ventral stress fiber adhesions (vsf FA, yellow box). Bar, 5 um. (**D**) Representative example of a fluorescence-recovery-after-photobleaching (FRAP) experiment performed on a GFP-actin expressing cell at a ventral stress fiber (vsf) region close to a focal adhesion. Yellow box indicates the photobleached region and the orange arrowhead the distal end of the ventral stress fiber. Scale, 3 um. (**E**) Quantification of the recovery speed (µm/min) for GFP-actin from focal adhesions located at the tips of dorsal stress fibers (dsf) or ventral stress fibers (vsf). Means +/- SEM, n (dorsal stress fibers) = 11; n (ventral stress fibers) = 17. (**F**) Activation of PA-GFP-actin in focal adhesions is followed by centripetal flow of photoactivated actin along the dorsal stress fiber. In contrast, PA-GFP-actin activated at a focal adhesion located at the tip of ventral stress fiber does not distribute from the adhesion to the stress fiber. Activated PA-GFP-actin is in green, focal adhesion marker mCherry-zyxin in red and the yellow dashed lines show the borders of the photoactivated region. Bar, 2,5 µm.

*Figure 2 continued on next page*

*Figure 2 continued*

The following figure supplements are available for figure 2:

**Figure supplement 1.** Alignment of focal adhesions linked to the ends of transverse arcs.

**Figure supplement 2.** Actin dynamics in focal adhesions located at the tips of dorsal and ventral stress fibers.

**Figure supplement 3.** Inhibition of myosin light chain phosphorylation results in formation of abnormally long dorsal stress fibers.

**Figure supplement 4.** Expression of dominant inactive Rif leads to abnormal elongation of dorsal stress fibers.

actin filament bundle (with a rate of ~0,26 µm/min) from focal adhesions located at the distal ends of dorsal stress fibers was observed (*Hotulainen and Lappalainen, 2006*). Importantly, when a FRAP analysis was performed on a corresponding ventral stress fiber region, only very slow elongation (~0,02 µm/min) of a bright actin filament bundle from the adhesion was observed. Instead, we mainly detected recovery of GFP-actin fluorescence evenly along the photobleached region (*Figure 2C-E*). As an alternative approach, we utilized photoactivatable (PA)-GFP-actin to follow its incorporation into dorsal and ventral stress fibers. In both cases, significant fraction of activated PA-GFP-actin remained at/close to focal adhesions, probably corresponding to actin filament pools associated with focal adhesions (*Tojkander et al., 2011*). Importantly, PA-GFP-actin displayed centripetal flow along the actin filament bundle from focal adhesions in dorsal stress fibers, while similar flow of PA-GFP-actin was not detected from focal adhesions located at the tips of ventral stress fibers (*Figure 2F*). Therefore, in contrast to dorsal stress fibers, ventral stress fibers do not elongate through vectorial actin polymerization at focal adhesions.

To elucidate whether inhibition of vectorial actin polymerization in focal adhesions at the tips of ventral stress fibers is dependent on tension applied by myosin II, we examined the morphology of the stress fiber network in cells treated with myosin light chain kinase (MLCK) inhibitor ML-7. This compound induced rapid disassembly of most contractile ventral stress fibers and transverse arcs, without affecting integrity of non-contractile dorsal stress fibers (*Figure 2—figure supplement 3C*). Importantly, dorsal stress fibers in cells treated for 2 h with ML-7 were ~1.5 times longer

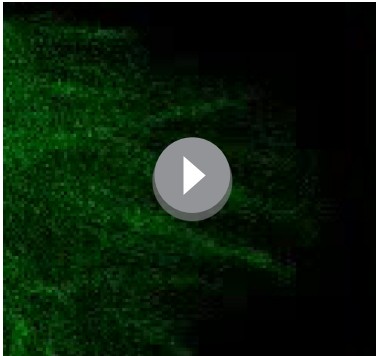

**Video 1.** Fluoresence recovery after photobleaching (FRAP) of GFP-actin at dorsal stress fiber-associated focal adhesions (FAs). GFP-actin at FAs of dorsal stress fibers were bleached with 100% laser power of 488 laser line for 1 ms. Signal of GFP-actin displayed relatively uniform recovery at the site of focal adhesion. Duration of the movie is 8,5 min and the display rate is 10 frames/second.

**Video 2.** Fluoresence recovery after photobleaching (FRAP) of GFP-actin at ventral stress fiber-associated focal adhesions (FAs). GFP-actin at FAs of dorsal stress fibers were bleached with 100% laser power of 488 laser line for 1 ms. Signal of GFP-actin displayed relatively uniform recovery at the site of focal adhesion. Duration of the movie is 8,5 min and the display rate is 10 frames/second.

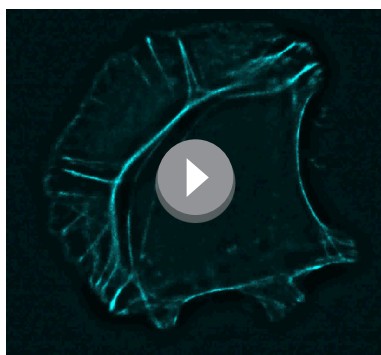

**Video 3.** Control movie on stress fiber dynamics in GFP-actin expressing U2OS cell. U2OS cells were transfected with GFP-actin 24 hr before imaging. Images were acquired every 15 s. Display rate is 15 frames/second and total duration is 73,3 min.

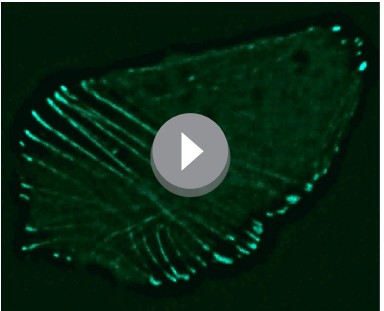

**Video 4.** Expression of dominant-inactive Rif causes uncontrolled dorsal stress fiber elongation. U2OS cells were transfected with Rif-TN and GFP-actin 24 hr prior to imaging. Loss of proper contractile structures due to Rif-TN expression causes abnormal elongation of the dorsal stress fibers as visualized by GFP-actin. Images were acquired every 15 s. Display rate is 15 frames/s, total video duration 53,5 min.

compared to the ones in control cells (*Figure 2—figure supplement 2D*). Similarly, disruption of contractile stress fibers by ROCK inhibitor, Y27632, or by over-expression of dominant inactive Rif GTPase (Rif-TN), which prevents assembly of contractile arcs (*Tojkander et al., 2011*), led to formation of abnormally long dorsal stress fibers (*Figure 2—figure supplement 3A and B*, and *Figure 2—figure supplement 4*). Importantly, live-imaging of GFP-actin expressing cells revealed that the abnormally long dorsal stress in Rif-TN transfected cells continued to elongate throughout the entire observation period. During their uncontrolled elongation, the dorsal stress fibers of Rif-TN expressing cells occasionally bent or fused with another elongating dorsal stress fiber initiated from the opposite side of the cell (*Figure 2—figure supplement 4A*; *Videos 3* and *4*).

To more directly test the role of myosin II-derived tension in stress fiber elongation, we examined whether local relaxation of contractile ventral stress fibers could re-induce vectorial actin polymerization at focal adhesions. Thus, we applied pointed laser ablation on ventral stress fibers (see *Figure 3—figure supplement 1*) followed by a similar FRAP assay as shown in *Figure 2D*. Whereas vectorial actin polymerization in intact contractile fibers was very slow (~0,02 μm/min), ablated ventral stress fibers displayed approximately 10-fold higher rate of vectorial actin polymerization (~0,23 μm/min), which is comparable to the one of dorsal stress fibers (*Figure 3A and B*, and data not shown). Importantly, also photoactivation experiments on PA-GFP-actin expressing cells demonstrated specific elongation of laser ablated ventral stress fibers and lack of vectorial actin polymerization at the focal adhesions located at the ends of non-ablated ventral stress fibers within the same cell (*Figure 3C and D*).

These data demonstrate that vectorial actin filament assembly, which promotes elongation of dorsal stress fibers, is inhibited in focal adhesions located at the tips of contractile ventral stress fibers. Furthermore, laser ablation experiments as well as assays with MLCK and ROCK inhibitors, and dominant inactive Rif provide evidence that tension applied by myosin II–mediated contractility is important for inhibition of vectorial actin polymerization at focal adhesions located at the ends of ventral stress fibers.

## Actin polymerization in focal adhesions is controlled by phosphorylation of VASP

Two proteins promoting actin filament elongation, Dia1 formin and vasodilator-stimulated phosphoprotein (VASP), have been linked to actin polymerization in focal adhesions (*Hotulainen and Lappalainen, 2006*; *Oakes et al., 2012*; *Watanabe et al., 1999*; *Gateva et al., 2014*; *Figure 4—figure supplement 1A and B*). Because from these proteins only VASP, and its family members Mena and Evl, accumulate to focal adhesions (*Reinhard et al., 1992*; *Gertler et al., 1996*; *Lambrechts et al., 2000*; *Hoffman et al., 2006*), we focused on examining the possible role of VASP in tension-controlled actin filament assembly in focal adhesions. Previous studies demonstrated zyxin-mediated

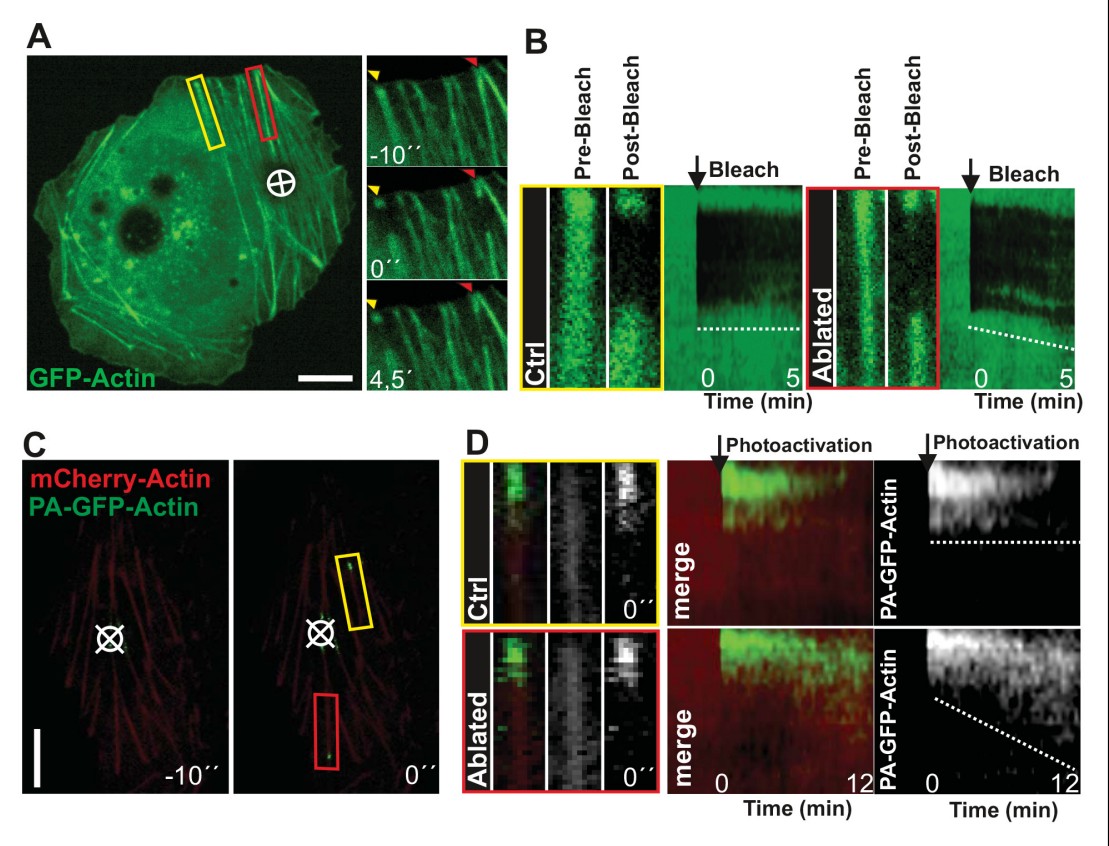

**Figure 3.** Local relaxation of ventral stress fibers induces vectorial actin polymerization at focal adhesion. (A) The effect of tension on actin polymerization at focal adhesions was monitored by fluorescence recovery after photobleaching (FRAP) in laser-ablated (indicated by red box/arrowhead) and intact ventral stress fibers (indicated by yellow box/arrowhead). FRAP experiment was initiated 10 seconds after ablation. Symbol (¤) indicates the ablation site. (B) Kymographs recorded along the center of ablated and non-ablated ventral stress fiber regions (shown in the red and yellow boxes to the left from the kymograps) reveal that non-ablated and ablated fibers display differences in actin dynamics. In contractile control fibers, the rate of vectorial actin polymerization is slow (0,023 µm/min +/- 0,007 µm/min; SEM; n = 17), whereas relaxation induces vectorial actin polymerization from the adhesion located at the end of ablated ventral stress fiber (0,257 µm/min +/- 0,035 µm/min; SEM; n = 12). See also *Figure 7B* for a graphical representation of the data. (C) Photoactivation of GFP-PA-actin in contractile (yellow box) and relaxed (red box) ventral stress fibers. (D) Kymograph analysis performed along the center of indicated ventral stress fiber regions (shown in yellow and red boxes to the left from the kymographs) demonstrate induction of vectorial actin polymerization at the focal adhesion located in the end of an ablated ventral stress fiber. However, no detectable vectorial actin polymerization occurred in the non-ablated ventral stress fiber. Photoactivation was performed 10 s after ablation of the contractile stress fiber. Bars, 10 µm.

The following figure supplement is available for figure 3:

**Figure supplement 1.** Method for laser ablation of ventral stress fibers.

recruitment of VASP to the sites of stress fiber repair and remodelling (*Smith et al., 2010*; *Hoffman et al., 2012*) and mechanosensitive recruitment of VASP to epithelial zonula adherens (*Leerberg et al., 2014*). However, whether the activity of VASP within adhesions can be regulated through tension has not been reported.

Immunofluorescence microscopy revealed that VASP localizes to focal adhesions located at the tips of both dorsal and ventral stress fibers (*Figure 4A and B*). Therefore, regulation of VASP localization does not offer an explanation for the lack of vectorial actin polymerization at the tips of ventral stress fibers. Interestingly, previous work demonstrated that phosphorylation of specific residues (Ser239 and Thr278) of VASP inhibit its actin filament binding and polymerization activities (*Harbeck et al., 2000*; *Benz et al., 2009*; *Figure 4C*). To study the possible role of VASP phosphorylation in actin filament assembly at focal adhesions, we first examined the localization of phosphorylated VASP in U2OS cells. From several VASP phospho-Ser239/Thr278 antibodies tested, only 16C2

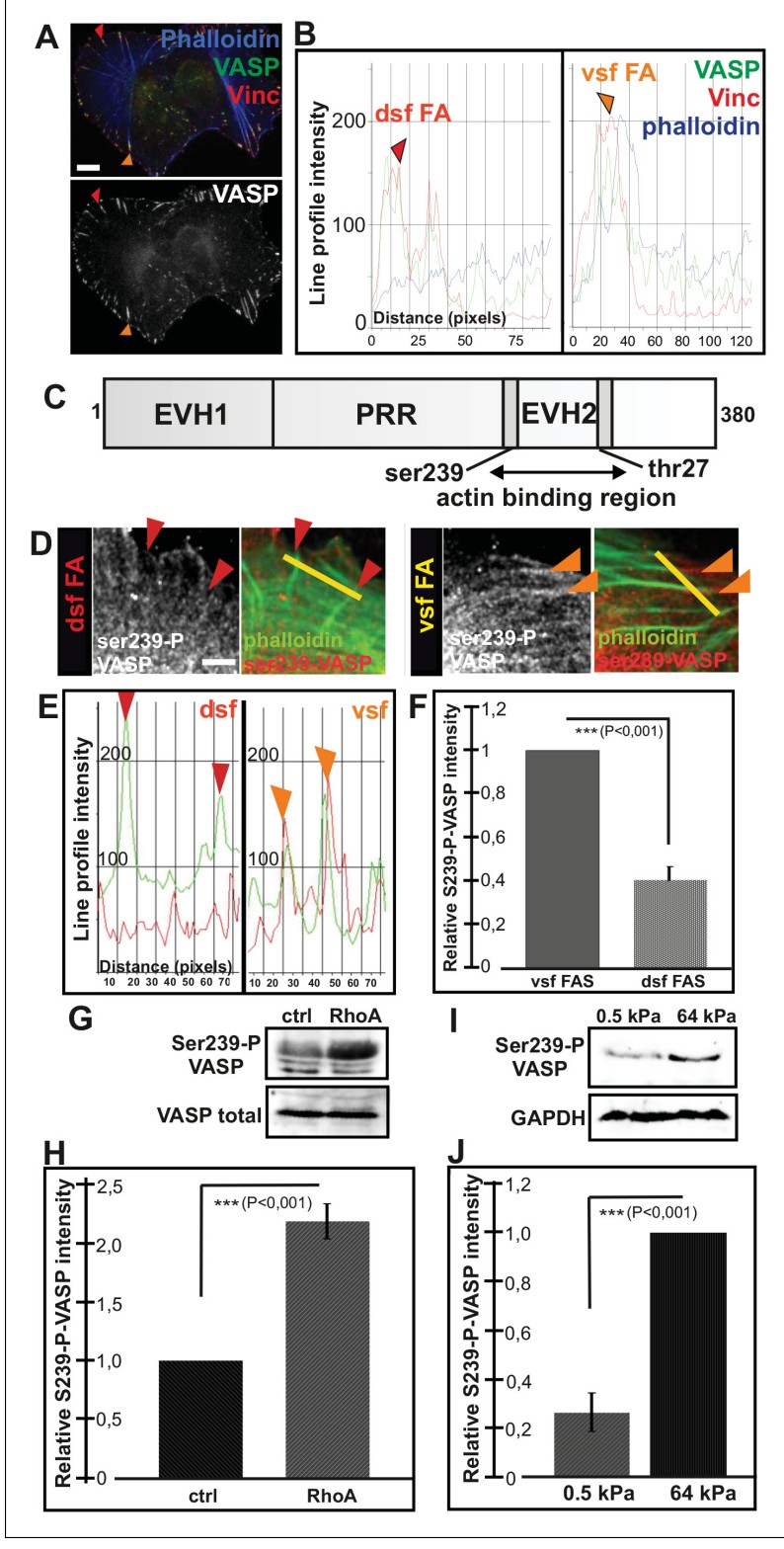

**Figure 4.** Increased VASP phosphorylation in focal adhesions located at the tips of ventral stress fibers. (**A**) VASP localizes to focal adhesions at the tips of both dorsal and ventral stress fibers. Focal adhesions positioned at the tips of dorsal (dsf FA) and ventral stress fibers (vsf FA) are indicated by red and orange arrowheads, respectively. Bar, 10 µm. (**B**) Line profile intensity graphs of the adhesions (highlighted in panel A) and adjacent stress fiber regions show similar localizations of VASP and vinculin in focal adhesions positioned at the tips of dorsal and

*Figure 4 continued on next page*

*Figure 4 continued*

ventral stress fibers. (**C**) The domain structure of VASP. Phosphorylation of VASP at Ser239 and Thr278 inhibits its actin polymerization activity. (**D**) Localization of phospho-Ser239 VASP in focal adhesions at the tips of dorsal (red arrowheads) and ventral stress fibers (orange arrowheads). Bar, 5 μm. (**E**) Line profile intensities along the yellow lines (indicated in panel D) demonstrating that phospho-Ser239 VASP is enriched at the tips of ventral stress fibers, but not at the tips of dorsal stress fibers. phospho-Ser239-VASP–red; actin-green (**F**) Quantification of the relative fluorescence intensity ratio of phospho-Ser239-VASP: total VASP in focal adhesions located at the tips of dorsal (dsf FAs) and ventral (vsf FAs) stress fibers. The obtained intensity value from ventral stress fibers was set to 1. Mean intensity values (+/- SEM) of 21 adhesions are shown. (**G**) Western blot analysis demonstrating that expression of dominant active Rho-14V expression leads to an increase in the total Ser239 phosphorylation levels of VASP. (**H**) Quantification of the relative phospho-Ser239-VASP: total VASP ratios in control and Rho-14V transfected cells. The ratio in control cells was set to 1 and the mean values (**+/-** SEM) from three separate experiments are shown. (**I**) Western blot analysis demonstrating increased phospho-Ser239-VASP levels in cells grown on stiff (64 kPa) martix compared to cells grown on compliant (0.5 kPa) matrix. (**J**) Quantification of the relative phospho-Ser239-VASP: total VASP ratios in cells grown on soft (0.5 kPa) or rigid (64 kPa) matrices. The phospho-Ser239-VASP: total VASP ratio in cells from stiff matrix was set to 1 and the mean values (**+/-** SEM) from three separate experiments are shown.

The following figure supplement is available for figure 4:

**Figure supplement 1.** VASP regulates the elongation of dorsal stress fibers.

(Millipore) worked in immunofluorescence experiments. Although the signal with this antibody was weak, it specifically stained focal adhesions located at the tips of ventral stress fibers, whereas enrichment of phospho-Ser239 VASP to adhesions at the tips of dorsal stress fibers was not detected (*Figure 4D, E and F*). To confirm this result by an alternative approach, we examined by Western blotting phospho-Ser239 and phospho-Thr278 VASP levels in control cells and in cells where the assembly of contractile ventral stress fibers was stimulated or inhibited by over-expression of dominant active RhoA or by plating cells on compliant matrix, respectively. In line with the data presented above, both phospo-Ser239 (*Figure 4G and H*) and phospho-Thr278 (data not shown) levels were >2-fold elevated in the cell population transfected with a construct expressing dominant active RhoA, whereas phospo-Ser239 levels were ~5-fold diminished in cells plated on soft matrix and unable to form contractile ventral stress fibers (*Figure 4I and J*). Thus, VASP phosphorylation in focal adhesions correlates with increased contractility of stress fibers.

VASP phosphorylation at Ser239 and Thr278 is regulated by cAMP- and cGMP dependent protein kinases PKA and PKG as well as by AMP-activated Protein Kinase (AMPK) (*Butt et al., 1994*; *Blume et al., 2007*). To elucidate the possible role of VASP phosphorylation in controlling actin polymerization in focal adhesions, we examined the effects of PKA, PKG and AMPK inhibitors (KT5720, DT-2, compound C and KT5823) on the organization of the stress fiber network. As AMPK inhibitors, compound C and KT5823, had most pronounced effects on the elongation of stress fiber precursors, we decided to focus on AMPK rather than PKA and PKG in this study.

Both of compound C and KT5823 inhibited VASP phosphorylation at Ser239 and Thr278 (*Figure 5C*). Importantly, incubation of U2OS cells for 4 hr in the presence of these inhibitors resulted in a nearly complete lack of contractile ventral stress fibers and defects in arc fusion. Furthermore, these inhibitors promoted formation of abnormally long dorsal stress fibers, which often bent at their proximal regions (*Figure 5A and B*). It is important to note that these inhibitor treatments also led to an increase in the total cell area that may result from the lack of contractile ventral stress fibers, which are important regulators cell morphogenesis.

To confirm that the stress fiber phenotype in compound C and KT5823 –treated cells was specific to VASP, and did not result from diminished phosphorylation of other AMPK targets, morphology of the stress fiber network of U2OS cells expressing a 'constitutively active' Ser239Ala;Thr278Ala VASP mutant was examined (*Benz et al., 2009*). Also Ser239Ala;Thr278Ala mutant VASP expressing cells displayed significantly longer dorsal stress fibers as compared to wild-type VASP expressing cells (*Figure 5D and E*). Furthermore, over-expression of Ser239Ala;Thr278Ala mutant VASP occasionally resulted in formation of 'curly' actin filament bundles (*Figure 4—figure supplement 1C*). Thus, inhibition of VASP phosphorylation at Ser239 and Thr278 results in a similar phenotype compared to

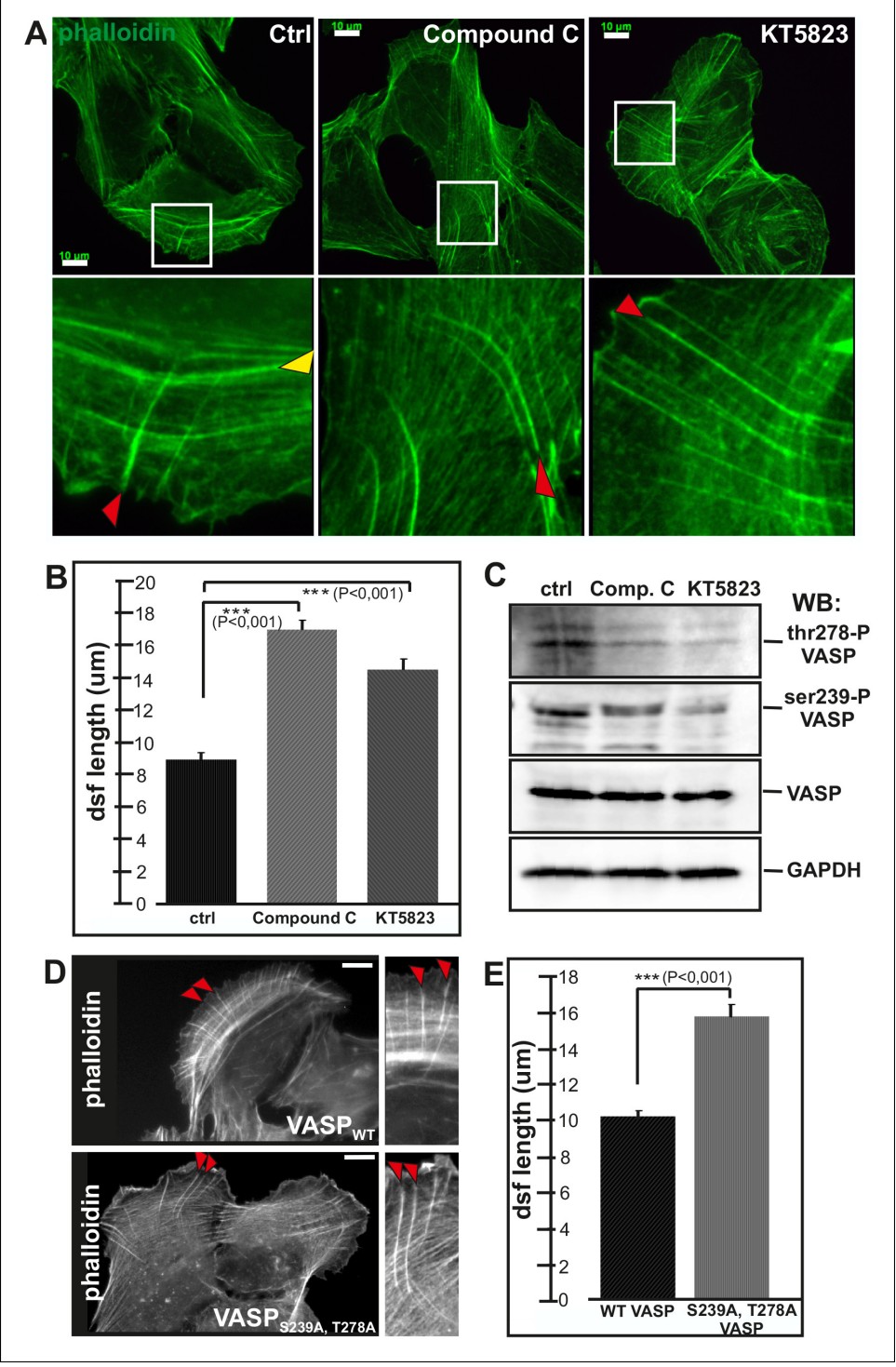

**Figure 5.** Elongation of dorsal stress fibers is regulated by VASP phosphorylation. (**A**) U2OS cells treated with cAMPK and PKA inhibitors, compound C and KT5823, display abnormally long dorsal stress fibers (red arrows). Also fusion of arcs appears defective in compound C and KT5823 treated cells, because thick actin bundles (yellow arrowhead in the control cell) are largely absent from these cells. Actin filaments were visualized by phalloidin. Bar, 10 μm. (**B**) Quantification of dorsal stress fiber lengths (μm) from control cells and cells treated with compound C or KT5823. Mean lengths (+/- SEM) of 40 dorsal stress fibers from each sample are shown. (**C**) Western blot demonstrating that lysates of compound C or KT5823 -treated cells display decreased phosphorylation of VASP at Ser239 and Thr278. GADPH and total VASP were probed as loading controls. (**D**) Expression of constitutively active VASP mutant (Ser239Ala;Thr278Ala) induces formation of abnormally long dorsal stress fibers, whereas

*Figure 5 continued on next page*

*Figure 5 continued*
similar phenotype was not observed in wild-type VASP expressing cells. Bar, 10 μm. (**E**) Quantification of dorsal stress fiber lengths from cells expressing wild-type and Ser239Ala;Thr278Ala mutant VASP. Mean lengths (+/- SEM) of 55 dorsal stress fibers measured from both samples are shown.

the one resulting from the inhibition of contractile arc assembly through over-expression of Rif-TN (see *Figure 2—figure supplement 4*), suggesting that VASP phosphorylation has a key role in tension-controlled actin filament assembly at focal adhesions.

Conversely, activation of AMPK by AICAR (*Carling et al., 2008*) leads to an increased Ser239 phosphorylation of VASP and early maturation of ventral stress fibers (*Figure 6A and B*). Cells exposed for 16 h to AICAR displayed shorter dorsal stress fibers compared to control cells and their ventral stress fibers were typically located close to cell perimeter (*Figure 6A*). This phenotype was similar to the one resulting from depletion of VASP (*Figure 4—figure supplement 1*), suggesting that AICAR indeed affects stress fibers mainly through inducing VASP phosphorylation. Importantly, activation of AMPK and VASP phosphorylation by AICAR treatment was sufficient to induce the formation of ventral stress fibers on soft (0.5 kPa) matrix, where their formation is normally inhibited (*Figure 6C and D*).

To directly test the role of VASP in mechanosensitive actin filament assembly at focal adhesions, we performed photoablation experiments on VASP knockdown cells, and applied FRAP to compare the vectorial actin polymerization rates of ventral stress fibers in control vs. VASP-depleted cells. These experiments revealed approximately 3-fold decrease in ablation-induced vectorial actin polymerization at the tips of ventral stress fibers in VASP knockdown cells compared to control cells (*Figure 7*), demonstrating that VASP is indeed important for mechanosensitive actin filament assembly in focal adhesion. Together, these results provide evidence that AMPK-mediated phosphorylation of VASP is essential for tension-sensitive inhibition of vectorial actin filament assembly at focal adhesions, and consequent formation and stabilization of ventral stress fibers.

## ADF/cofilin-mediated disassembly of non-contractile actin filament bundles is important for maturation of ventral stress fibers

Maturing arcs are typically connected to several dorsal stress fibers and focal adhesions, but only the ones located at the ends of the arc bundle are used for the formation of a ventral stress fiber (*Figure 2—figure supplement 1*). To gain insight into the fate of other arc-associated dorsal stress fibers and focal adhesions, we performed live-imaging of cells expressing GFP-zyxin and mCherry-actin. Focal adhesions connected to the ends of arcs elongated during maturation of arcs into a ventral stress fiber, whereas adhesions associated with the central region of the arc through dorsal stress fibers diminished in size and eventually disappeared (*Figure 8—figure supplement 1*). To reveal what happens to those dorsal stress fibers, which are located at the 'unstable zone' at the central region of the leading edge (see *Figure 8D*), we followed the stress fiber network in cells expressing GFP-actin. These experiments revealed that dorsal stress fibers, oriented perpendicularly to the contractile arc, sense weaker myosin II -generated tension and disassemble during stress fiber maturation process (*Figure 8A and C*). Furthermore, those dorsal stress fiber regions, which reach beyond the contractile arc/ventral stress fiber, disassemble during the process (*Figure 8B*).

Actin depolymerizing factor (ADF)/cofilin proteins are essential regulators of F-actin disassembly in all eukaryotic cells (*Poukkula et al., 2011*). Interestingly, ADF/cofilins were recently shown to preferentially bind and disassemble flexible actin filaments in vitro. ADF/cofilins did not localize to contractile stress fibers in intact cells, but translocated to stress fibers when pre-stretched elastic substratum was relaxed (*Hayakawa et al., 2011*). Furthermore, ADF/cofilins affect the actin filament bending mechanics (*McCullough et al., 2008*; *Elam et al., 2013*). Thus, we examined whether ADF/cofilins could be responsible for specific disassembly of those dorsal stress fibers that are not under tension in migrating U2OS cells. The major ADF/cofilin isoform, cofilin-1, is highly abundant protein in non-muscle cells where it displays mainly diffuse cytoplasmic localization. Interestingly, endogenous cofilin-1 and flag-tagged cofilin-1 also localized to dorsal stress fibers in U2OS cells, whereas contractile ventral stress fibers and thick arcs did not exhibit detectable enrichment of cofilin-1 (*Figure 9A and B*; *Figure 9—figure supplement 1A, B and C*). Furthermore, cofilin-1 localized to

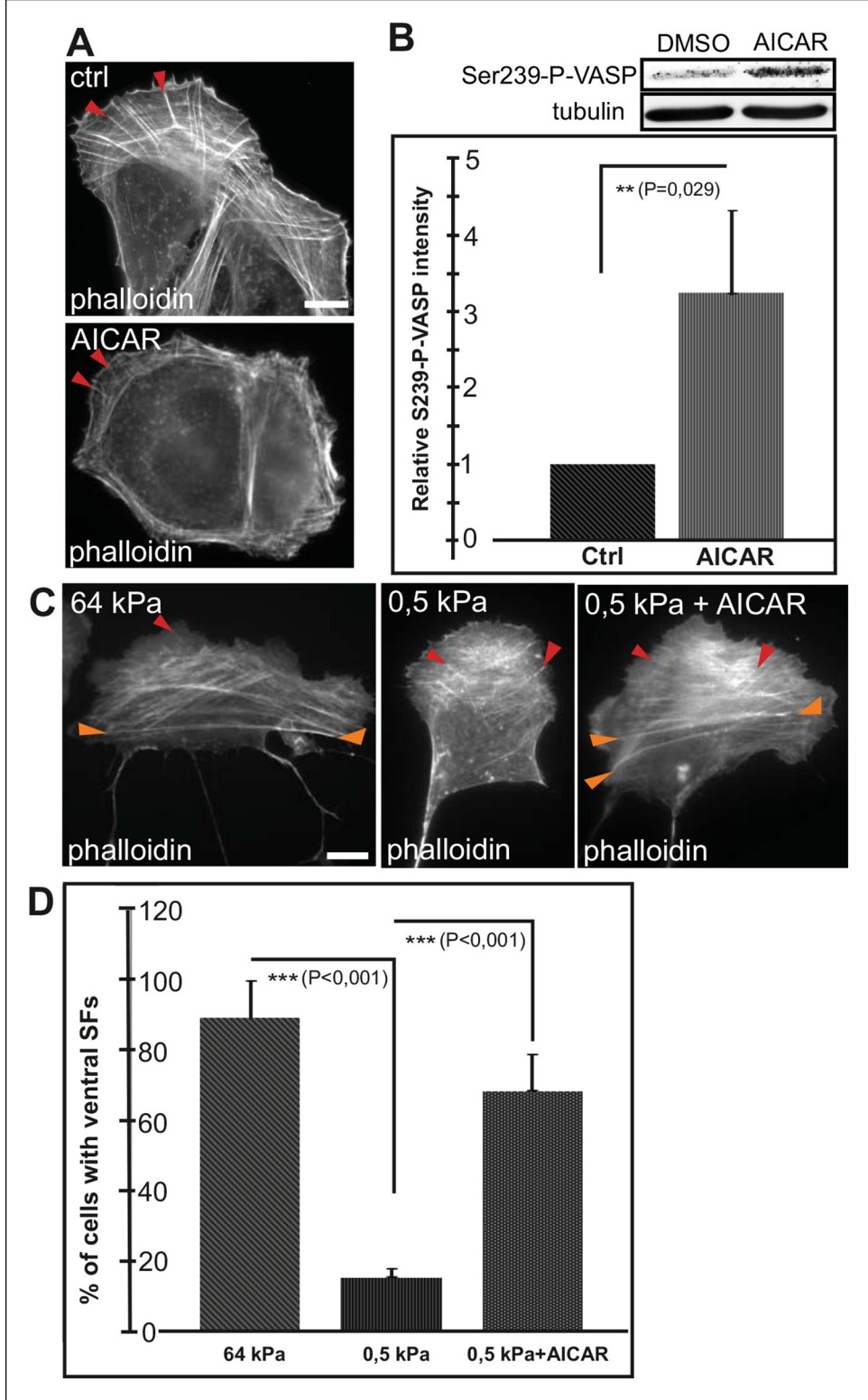

**Figure 6.** AMPK activation leads to maturation of contractile actomyosin bundles. (**A**) U2OS cells grown on glass and treated with AMPK activator AICAR exhibited short dorsal stress fibers. Ventral stress fibers were typically localized close to cell edges, indicating early maturation of the contractile actomyosin bundles compared to control cells. (**B**) Western blot analysis of the corresponding samples showed elevation of Ser239-P-VASP in AICAR treated cells. A representative Western blot and quantification of three separate experiments (mean +/- SEM) are shown. (**C**) Activation of AMPK bypasses the need for stiff substrata in ventral stress fiber formation. U2OS cells

*Figure 6 continued on next page*

*Figure 6 continued*

grown on soft (0.5 kPa) matrix do not typically contain contractile ventral stress fibers, but their formation on soft matrix can be induced by AICAR. (**D**) Quantification of the amount of cells (%) containing ventral stress fibers. For each condition, 50–70 cells were analysed and the data are presented as mean +/- SEM.

the 'curly' actin filament bundles that were occasionally present in cells over-expressing Ser239Ala; Thr278Ala mutant VASP (*Figure 9—figure supplement 1D*). These bundles contain myosin II, but exert defective contractile properties as detected by live cell imaging (data not shown). Thus, cofilin-1 appears to localize specifically to dorsal stress fibers and other non-contractile actin bundles in U2OS cells.

Depletion of cofilin-1 leads to defects in multiple actin-based structures such as lamellipodia and sites of endocytosis due to diminished filament disassembly as well as to cortical F-actin accumulation due to excessive myosin II activity (e.g. *Hotulainen et al., 2005*; *Sidani et al., 2007*; *Kiuchi et al., 2007*; *Wiggan et al., 2012*). Importantly, depletion of cofilin-1 from U2OS cells resulted also in an appearance of abnormally thick and long dorsal stress fibers (*Figure 9C and D*; *Figure 9—figure supplement 1E*). Furthermore, fusion of arcs during their centripetal flow was diminished, leading to problems in the formation of proper ventral stress fibers (*Figure 9—figure supplement 1F*; *Video 5*). Defects in arcs fusion suggest that proper ADF/cofilin-mediated turnover

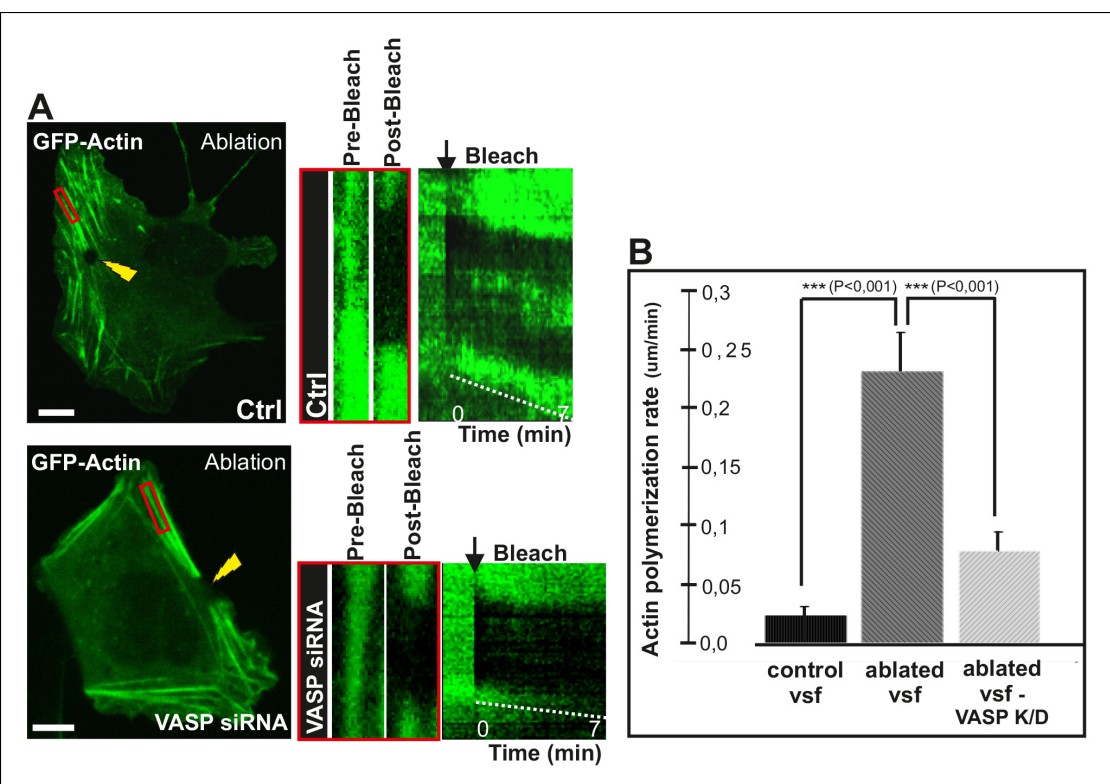

**Figure 7.** VASP-depletion leads to a decrease in tension-sensitive actin polymerization at focal adhesions. (**A**) Actin polymerization at focal adhesions was monitored by fluorescence recovery after photobleaching (FRAP) in laser-ablated ventral stress fibers of control cells and VASP knockdown cells. FRAP experiments were initiated 10 seconds after ablation. Yellow arrowheads indicate the ablation sites and red boxes the regions of stress fibers that were followed for vectorial actin polymerization. Kymographs on the right were recorded along the center of the ablated ventral stress fibers. Release of tension induces vectorial actin polymerization from the adhesion located at the end of an ablated ventral stress fiber in a control cell, whereas the rate of vectorial actin polymerization was slower in an ablated ventral stress fiber in a VASP-depleted cell. Bar, 10 um. (**B**) Quantification of vectorial actin polymerization rates in intact ventral stress fibers, in ablated ventral stress fibers of control cells, and in ablated ventral stress fibers of VASP-depleted cells. Mean +/- SEM is shown; n (intact ventral stress fibers) = 17; n (ablated ventral stress fibers) = 12; n (ablated ventral stress fibers from VASP knockdown cells) = 17.

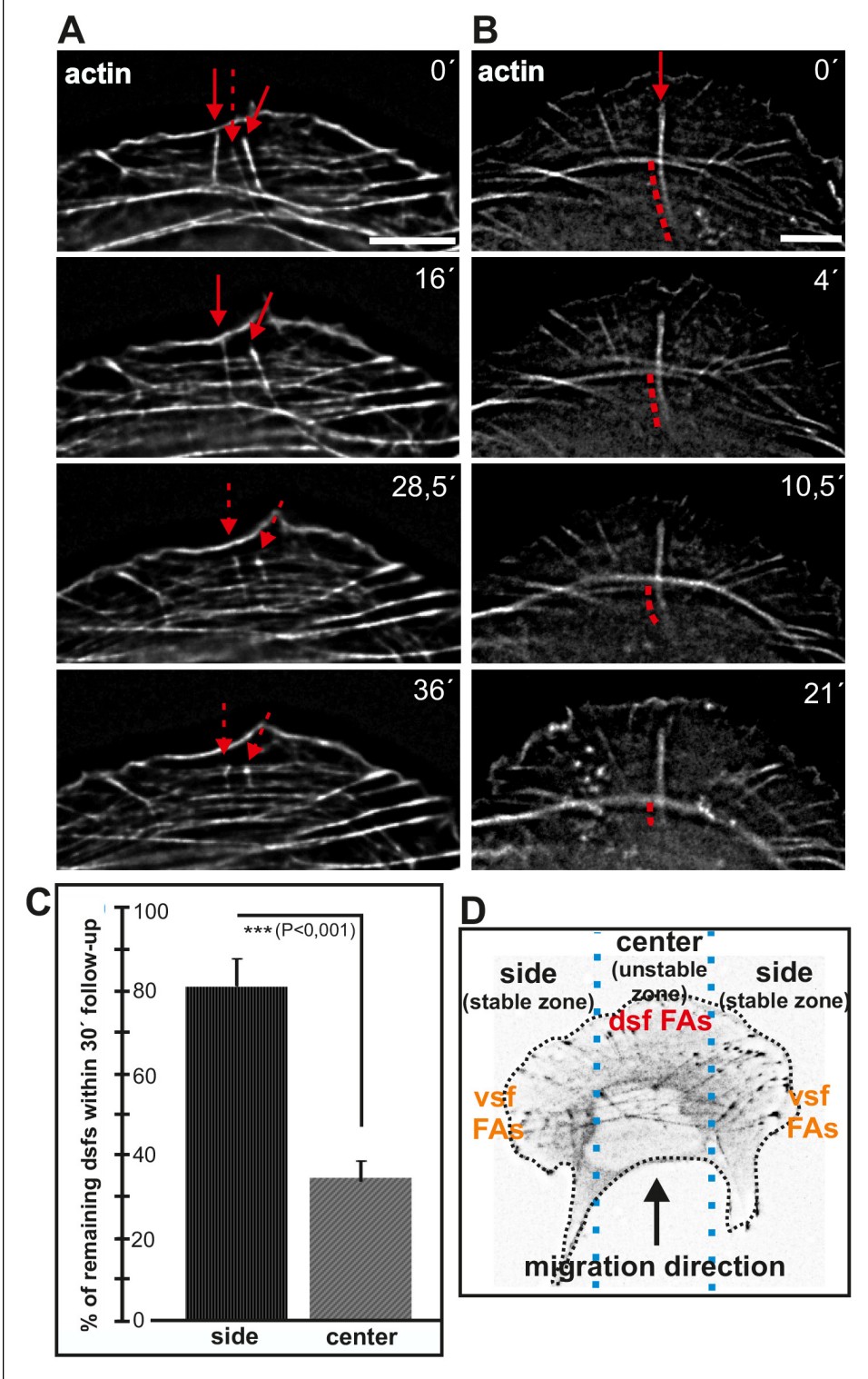

**Figure 8.** Dorsal stress fibers exhibit different lifespans depending on their interactions with the actomyosin network. (**A**) Individual frames from a representative movie of GFP-actin expressing cell demonstrating the disassembly of non-contractile dorsal stress fibers located at the 'unstable' zone. Bar, 5 µm. (**B**) Frames from a movie of GFP-actin expressing cell displaying the disassembly of the non-contractile dorsal stress fiber region extending beyond the contractile transverse arc. Bar, 5 µm. (**C**) Quantification of the stability of dorsal stress fibers at different cell regions revealed that these actin bundles are more stable at the sides of the leading edge as compared to the central region of the leading edge. Amount of dorsal stress fibers (%), remaining after 30 min
*Figure 8 continued on next page*

*Figure 8 continued*
follow-up, is shown (mean +/- SD), n = 5 cells, 8–20 fibers per cell were analysed. (**D**) Representation of the 'unstable' and 'stable' dorsal stress fiber zones in U2OS cells.
The following figure supplement is available for figure 8:

**Figure supplement 1.** Enlargement and lifespan of focal adhesions depends on their location in migrating cells.

of dorsal stress fibers is required for proper coalescence of dorsal stress fiber–associated arcs. Together, these data provide evidence that ADF/cofilins are important for turnover of non-contractile dorsal stress fibers, whereas contractile arcs and ventral stress fibers appear to be protected from ADF/cofilin-mediated F-actin disassembly.

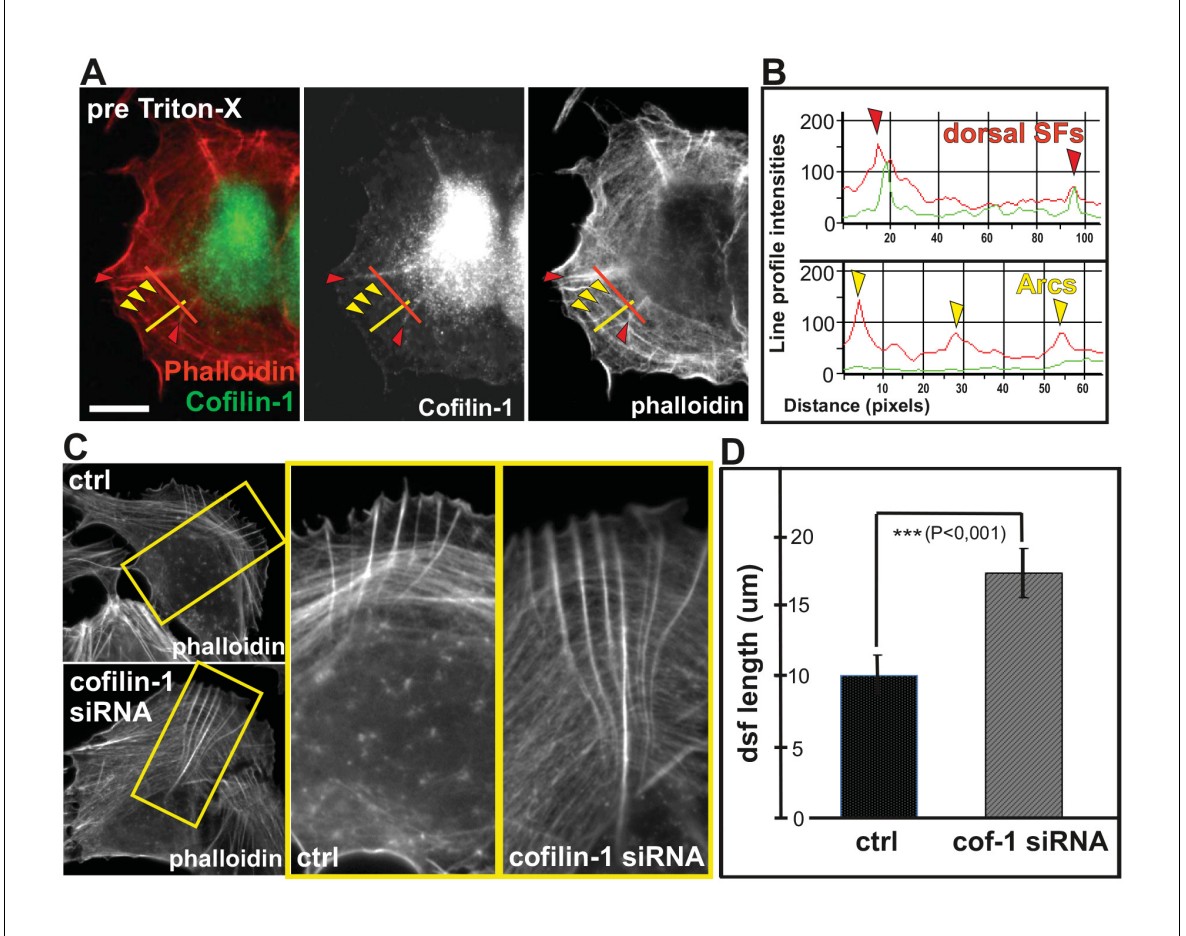

**Figure 9.** Cofilin-1 promotes disassembly of non-contractile dorsal stress fibers. (**A**) Endogenous cofilin-1 localizes to dorsal stress fibers (red arrowheads) but it is absent from contractile arcs (yellow arrowheads) as shown by phalloidin and anti-cofilin-1 staining of a U2OS cell treated with Triton-X 100 prior to PFA fixation. Bar, 10 μm. (**B**) Line intensity profiles show incorporation of cofilin-1 into dorsal stress fibers (dsf) but not to the contractile ventral stress fibers (vsf). Cofilin (green); Actin (red). (**C**) Depletion of cofilin-1 leads to an appearance of abnormally long dorsal stress fibers and defects in the fusion of transverse arcs. Bar, 10 μm. (**D**) Quantification of the lengths of dorsal stress fibers (μm) in control and cofilin-1-depleted cells. Mean lengths (+/- SEM) of 50 dorsal stress fibers from control and cofilin-1 RNAi cells are displayed in the graph.
The following figure supplement is available for figure 9:

**Figure supplement 1.** Cofilin-1 localizes to dorsal stress fibers and affects their turnover.

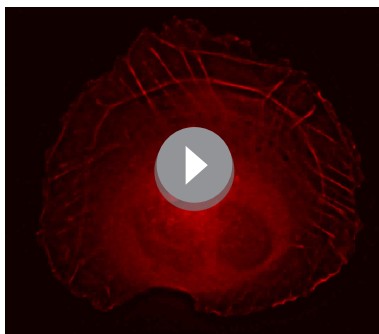

**Video 5.** Effects of cofilin-1 depletion on stress fiber dynamics. Cofilin-1 depleted U2OS cells were transfected with GFP-actin 24 h prior to imaging. Lack of cofilin-1 leads to elongated dorsal stress fibers and impairs transverse arc fusion. Images were acquired every 15 s. Display rate is 15 frames/s, total video duration 62,5 min.

## Discussion

Ventral stress fibers play an important role in cell adhesion, morphogenesis and migration, but how these and other contractile actomyosin bundles are generated has remained elusive. Here we have revealed several new aspects concerning the mechanisms underlying the assembly of contractile ventral stress fibers. We provide evidence that: (1) Formation of ventral stress fibers from their precursors (arcs and dorsal stress fibers) is a mechanosensitive process. (2) Arcs fuse with each other during centripetal flow to form thicker and more contractile actomyosin bundles, which apply tension to focal adhesions located at their ends. (3) This tension activates AMPK-mediated phosphorylation of VASP that leads to inhibition of vectorial actin polymerization at focal adhesions. (4) AMPK-mediated VASP phosphorylation is necessary for assembly and proper alignment of contractile ventral stress fibers. Conversely, activation of AMPK can bypass the need of stiff matrix for ventral stress fiber assembly. (5) ADF/cofilin–mediated disassembly of non-contractile dorsal stress fibers is important for the proper maturation of the stress fiber network. We propose that similar mechanosensitive actin filament assembly and disassembly may have general role in formation and alignment of diverse contractile actomyosin bundles in different cell-types.

A working model for mechanosensitive assembly of contractile ventral stress fibers is presented in *Figure 10*. Nascent adhesions appear at the lamellipodium of migrating cell and a fraction of them matures to focal adhesions (*Burnette et al., 2011*; *Choi et al., 2008*). Dorsal stress fibers are initiated from focal adhesions located at the leading edge of cells, and elongate through 'vectorial' actin polymerization at focal adhesions, mediated at least by VASP and Dia1 formin (*Hotulainen and Lappalainen, 2006*; *Watanabe et al., 1999*; *Gateva et al., 2014*). Similarly to filopodia, where VASP localizes at the tip-complex and promotes assembly of unipolar actin filament bundles, VASP at focal adhesions is expected to catalyse polymerization of an unipolar actin filament bundle towards the cell center. Elongating dorsal stress fibers associate with multiple myosin II containing arcs, which are derived from the lamellipodial actin structures (*Figure 10A*). During centripetal flow of this spider-net like structure, arcs fuse with each other to generate a thicker, more contractile bundle (*Figure 10B and C*). As a result, those focal adhesions that are attached via dorsal stress fibers to the ends of the contractile arc sense strong myosin II-generated tension, leading to their enlargement and turning along the direction of the arc. In support to this, ventral stress fibers apply strong traction forces to the substrate through their terminally located focal adhesions (*Figure 2* and *Möhl et al., 2012*). Moreover, tension-mediated maturation of terminal focal adhesions leads to inhibition of vectorial actin polymerization, which is at least partially mediated by phosphorylation of VASP (*Figure 10C*). However, because VASP-depletion did not result in a compete inhibition of vectorial actin polymerization after releasing the tension in ventral stress fibers (*Figure 7*), other proteins are also likely to contribute to vectorial actin polymerization at focal adhesions. It is also important to note that, although the actin polymerization activity of VASP is inhibited at focal adhesions under high tension, VASP protein is still present in that location. Thus, VASP may contribute to integrity of the adhesions at the tips of ventral stress fibers through its other activities, including actin filament bundling (*Bear and Gertler, 2009*). Furthermore, VASP phosphorylation is not expected to inhibit all actin dynamics in focal adhesions, because VASP appears to specifically contribute to vectorial actin polymerization at focal adhesions, whereas other proteins such as formins may promote the turnover of other focal adhesion-associated actin filament populations.

Importantly, inhibition of vectorial actin polymerization is essential for proper alignment and contractility of the ventral stress fiber, because continuous elongation of the actin filament bundle from focal adhesions would counteract myosin II-driven shortening of the actomyosin bundle. Finally, we

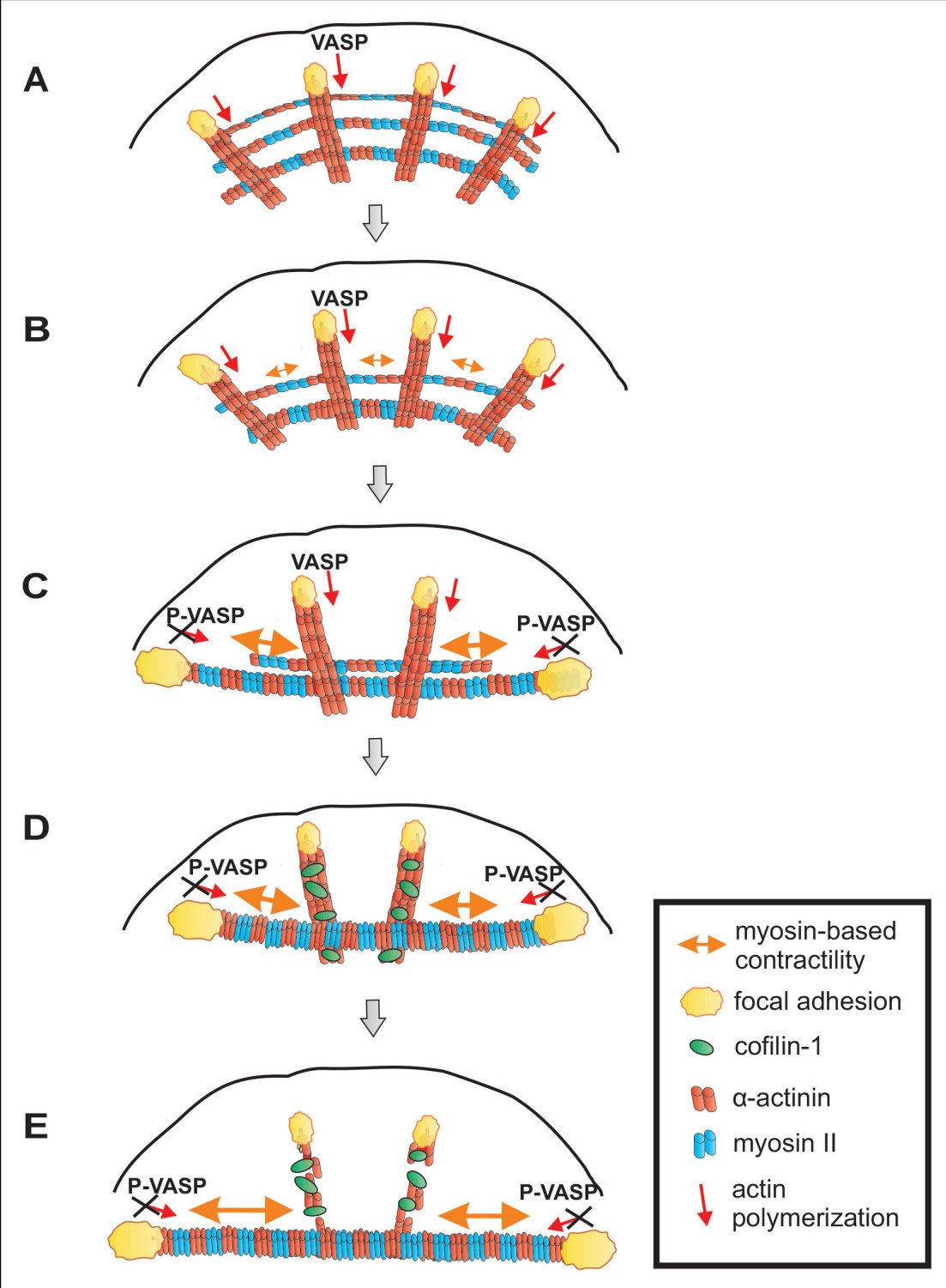

**Figure 10.** A working model for mechanosensitive generation of ventral stress fibers in U2OS cells. (**A**) Dorsal stress fibers elongate through vectorial actin polymerization from focal adhesions located at the leading edge of the cell, and form a spider net -like structure with multiple transverse arcs. At least Dia1 formin and VASP are involved in vectorial actin polymerization and consequent dorsal stress fiber elongation from focal adhesions. (**B**) Arcs flow along the elongating dorsal stress fibers towards the cell center, and fuse with each other to form thicker and more contractile actomyosin bundles. (**C**) Tension provided by the contraction of arcs is mediated through dorsal stress fibers to those focal adhesions that are linked to the end of the arc. This leads to enlargement of 'terminal' adhesions and their alignment along the direction of the contractile arc bundle. Tension provided by myosin II –driven contractility of the arc inhibits vectorial actin polymerization in 'terminal' focal adhesions, and this is at least partially mediated by

*Figure 10 continued on next page*

*Figure 10 continued*

VASP phosphorylation. Consequently, elongation of the actomyosin bundle ceases, thus allowing its efficient contractility. Please note that focal adhesions are likely to be composed of many actin filament populations, and for simplicity only the one undergoing vectorial actin polymerization and thus promoting stress fiber elongation is shown in the model. (D) Cofilin-1 specifically binds to and promotes the disassembly of non-contractile dorsal stress fibers, which are connected to the central regions of the arc and thus do not participate in the formation of the ventral stress fiber. (E) Whereas non-contractile stress fibers are disassembled by cofilin-1, contractile stress fibers are protected from tension-sensitive cofilin-1–induced severing. Eventually, this leads to the formation of a contractile ventral stress fiber, which is connected to one large focal adhesion at its each end and aligned perpendicularly to the direction of lamellipodium extension.

The following figure supplement is available for figure 10:

**Figure supplement 1.** Differential localizations of Tm1 and Tm5NM1 in focal adhesions at the tips of dorsal and ventral stress fibers.

propose that mechanosensitive binding of cofilin-1 to those dorsal stress fibers and dorsal stress fiber regions, which are not under myosin II-applied tension, leads to disassembly of these 'non-productive' regions of the stress fiber network. Consequently, the focal adhesions located at the distal ends of 'central' dorsal stress fibers, which are oriented perpendicularly to the contractile arcs and hence do not sense strong myosin II–derived tension, diminish in size and disappear. On the other hand, contractile arcs and mature ventral stress fibers are protected from ADF/cofilin-mediated actin filament disassembly (*Figure 10D and E*). Consistent with this model, dorsal stress fibers and arcs form in cells grown on soft substrata, whereas contractile ventral stress fibers fail to assemble in compliant matrix (*Figure 1A*).

Focal adhesions are mechanosensitive structures (e.g. *Iskratsch et al., 2014*). Their maturation from nascent adhesions and maintenance require relatively small forces that can be generated by retrograde actin flow without myosin II-driven contractility (*Oakes et al., 2012*; *Stricker et al., 2013*). However, focal adhesions mature into larger, elongated adhesions under stronger, myosin II-derived tension (*Geiger et al., 2009*). Myosin II -generated force also affects the protein composition and dynamics in focal adhesions (*Kuo et al., 2011*; *Schiller et al., 2011*; *Wolfenson et al., 2011*). Here, we provide evidence that vectorial actin polymerization, which drives elongation of stress fibers, is strictly controlled in focal adhesions. Thus, different force

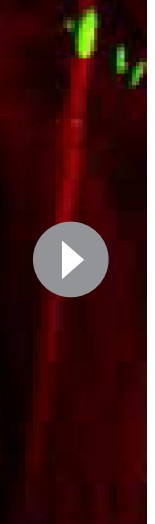

**Video 7.** Ablation of contractile ventral stress fibers does not affect the position of focal adhesions. An example of a laser-ablated ventral stress fiber from a U2OS cell co-expressing mCherry-Actin and GFP-Zyxin. Five captures, every 1 s, were taken before the ablation, and retraction of the ventral stress fiber was followed for 10 x 1 s before bleaching the region below the focal adhesion. Recovery of mCherry-Actin signal was detected at the bleached area of the relaxed fiber. Display rate is 10 frames/s, and the total duration of the video is 9 min. Please note, that position of the focal adhesion (indicated by GFP-Zyxin) is not significantly affected by ablation of the associated stress fiber.

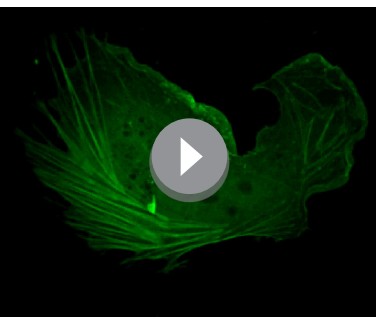

**Video 6.** VASP-depleted cells lack dorsal stress fibers and have poorly aligned contractile actomyosin bundles. VASP-depleted U2OS cells were transfected with GFP-actin one day prior to imaging. Images were acquired every 15 s. Display rate is 15 frames/s, total video duration 21,3 min.

regimes seem to have distinct effects on actin dynamics and molecular composition of focal adhesions. While weak forces exerted by retrograde actin flow appear to be required for VASP recruitment and to promote vectorial actin polymerization at focal adhesions, stronger force applied by contractility of the myosin II-containing ventral stress fiber efficiently inhibits actin polymerization at focal adhesions. Importantly, during this process VASP is phosphorylated by AMPK, whose activity at least in muscle cells can be controlled by tension through a currently uncharacterized mechanism (*Blair et al., 2009*). It is likely that activities or localizations of additional actin-polymerization associated proteins in focal adhesions are regulated by contractility. Indeed, an interaction partner of VASP, palladin (*Gateva et al., 2014*), as well as tropomyosins Tm1 and Tm5NM1 (*Figure 10—figure supplement 1*) are enriched in focal adhesions located at the tips of dorsal stress fibers, but absent from adhesions at the tips of ventral stress fibers. Furthermore, formins contribute to actin filament nucleation and/or processive polymerization in focal adhesions, and it is likely that also their activities are regulated during this process. Recent studies demonstrated that low-regime forces (<3 pN) increase the actin polymerization activity of formins (*Courtemanche, et al., 2013*; *Jégou et al., 2013*). Furthermore, the activity of formins can be controlled by a local increase in G-actin levels (*Higashida et al., 2013*). In the future, it will be interesting to examine the effects of imposed external forces, comparable in magnitude to tension applied by a myosin II-containing ventral stress fiber, on formins. However, similar to shown here for VASP, we propose that possible regulation of formin activity in focal adhesions is more likely controlled through biochemical signalling cascades than direct mechanical regulation of the formin molecule. This is because culturing cells on hyaluronic acid containing soft gels can produce a similar formation of ventral stress fibers that is otherwise observed only in cells cultured on rigid substrates (*Chopra et al., 2014*). Furthermore, several tyrosine kinases play an important role in focal adhesion mechanosensing, and their inactivation can shift the stiffness regime for assembly of large focal adhesions and ventral stress fibers (*Prager-Khoutorsky et al., 2011*).

In addition to vectorial actin polymerization at focal adhesions, actin filaments within the stress fiber network undergo turnover with a half-life of an approximately one minute (*Hotulainen and Lappalainen, 2006*). We propose that maintenance or disappearance of individual stress fibers depends on a balance between assembly and tension-sensitive disassembly of actin filaments. In addition to its role in actin polymerization in focal adhesions, VASP contributes to actin filament assembly within the stress fiber network (*Gateva et al., 2014*; *Smith et al., 2010*; *Hoffman et al., 2012*). Our data propose that stress fibers are maintained or become thicker under tension, whereas mechanosensitive binding of ADF/cofilins to stress fibers that are not under tensions shifts the balance from steady state (or net assembly) to net disassembly. Eventually, this leads to disappearance of the stress fiber and the focal adhesion associated with its end.

What are the functions of dorsal stress fibers? Our data demonstrate that arc fusion during centripetal flow occurs preferentially at the intersections with dorsal stress fibers. This suggests that dorsal stress fibers may functions as 'rails' to facilitate coalescence of adjacent arcs in the 3D-environment inside lamellum. However, arc fusion and formation of focal adhesion–attached ventral stress fibers can occur also in VASP-depleted cells, which either do not contain dorsal stress fibers or where these actin filament bundles are very thin and fragile (*Figure 4—figure supplement 1A and B*; *Video 6*). Furthermore, many cell-types including epithelial cells can assemble peripheral actomyosin bundles resembling ventral stress fibers in the apparent absence of dorsal stress fibers. Thus, focal adhesion–attached contractile actomyosin bundles can be generated at least in non-motile cells without prominent dorsal stress fibers. It is, however, important to note that the stress fiber network is typically poorly organized in VASP-depleted U2OS cells (*Figure 4—figure supplement 1B*), suggesting that dorsal stress fibers are required for proper alignment of ventral stress fibers in migrating cells. Furthermore, dorsal stress fibers play an important role in directional cell migration (*Kovac et al., 2013*).

Collectively, our findings reveal that mechanosensitive actin filament assembly and disassembly are essential for generation of contractile ventral stress fibers, and function as selection processes to ensure proper alignment of ventral stress fibers perpendicularly to the direction of cell migration. In the future, it will be important to identify the signalling pathway regulating mechanosensitive phosphorylation of VASP in focal adhesions. Here it will be especially interesting to examine the possible contribution of mechanosensitive $Ca^{2+}$ channels and $Ca^{2+}$-activated CaMKK family kinases, because the latter can activate AMPK kinase (*Carling et al., 2008*). Furthermore, it will be important to reveal

how the activities of other proteins contributing to actin polymerization at focal adhesions are regulated by myosin II -dependent tension. Finally, it will be interesting to examine how mechanosensitive actin filament assembly and disassembly contribute to generation and proper alignment of stress fibers and other contractile actomyosin bundles in the tissue environment.

## Materials and methods

### Cell culture

Human osteosarcoma (U2OS) cells were maintained as described in *Hotulainen and Lappalainen (2006)*. Transient transfections were performed with LipofectamineTM2000 (Invitrogen) according to manufacturer's instructions. Cells were subsequently incubated for 24 hr and further fixed with 4% PFA or detached with trypsin-EDTA and plated on fibronectin-coated (10 µg/ml fibronectin) glass-bottomed dishes (MatTek) for live cell imaging. Fibronectin-coated CytoSoft<sup>TM</sup> 35 mm biocompatible silicone dishes (Advanced BioMatrix) with elastic modulus of 0.5 and 64 kPa were used for studying the effect of matrix rigidity on stress fiber composition. For siRNA silencing, 2100 ng of pre-annealed $3'$ Alexa Fluor 488–labelled oligonucleotide duplexes were transfected into cells on 35 mm plates by using GeneSilencer's siRNA transfection reagent (Gene Therapy Systems) according to the manufacturer's instructions. Cells were incubated for 72–96 hr for efficient depletion of the target protein. For inhibition of VASP phosphorylation, cells were treated with AMPK inhibitor, Compound C (final concentration of 5 uM for 5 hr) or PKA/PKG inhibitor, KT5823 (1 uM, 4 hr) and for AMPK activation, 25 uM AICAR was used for 16 hr. For disruption of contractile structures, cells were treated with either myosin light chain kinase (MLCK) inhibitor ML-7 (1 uM, 2 hr) and ROCK inhibitor, Y27632 (1 uM, 2 hr). All chemical compounds were purchased from Sigma-Aldrich.

### Live cell imaging

Cells were transfected, incubated for 24 hr, and re-plated prior to imaging on 10 µg/ml fibronectin–coated glass-bottomed dishes (MatTek Corporation). The time-lapse images were acquired with 3I Marianas imaging system (3I intelligent Imaging Innovations), with an inverted spinning disk confocal microscope Zeiss Axio Observer Z1 (Zeiss) and a Yokogawa CSU-X1 M1 confocal scanner, or with an inverted microscope (IX-71; Olympus) equipped with a Polychrome IV monochromator (TILL Photonics). Both systems have appropriate filters, heated sample chamber (+37°C), and controlled $CO_2$. With 3I Marianas, a 63x/1.2 W C-Apochromat Corr WD = 0.28 M27 objective was used. SlideBook 5.0 software (3I intelligent Imaging Innovations) and sCMOS (Andor) Neo camera were used for the image acquirement and recording. With Olympus, a 60x water objective with $1.6\times$ magnification was used. TILL Vision 4 software (TILL Photonics) and Imago QE (TILL Photonics) and Andor iXon (Andor) cameras were used for the image acquirement and recording. Deconvolution of the time-lapse videos was performed with AutoQuant AutoDeblur 2D non-blind Deconvolution (AutoQuant Imaging, Inc.). Further analyses of the video frames were performed with Image Pro Plus 6.0.

### Traction force microscopy (TFM)

U2OS cells, transfected with Cherry-Actin, were cultured for 3–8 hr on collagen-1-coated polyacrylamide (PAA) gel substrates (elastic modulus = 26 kPa) that were coated with sulfate fluorescent microspheres (diameter = 100 or 200 nm, Life Technologies) (*Marinkovic et al., 2012*). Using an inverted fluorescence microscope (Leica DMI6000), images of cells and of the fluorescent microspheres directly underneath the cells were imaged during the experiment and after cell detachment with trypsin. By comparing the fluorescent microsphere images before and after cell detachment, we computed spatial maps of cell-exerted displacement. With knowledge of the displacement field and that of the substrate stiffness, we computed the traction field using the well-established method of constrained fourier transform traction microscopy (*Butler et al., 2002*; *Krishnan et al., 2009*). From the cell traction map, we computed local force within an ~13µm<sup>2</sup> area around pre-selected points corresponding to tips of focal adhesions at either dorsal (red) or ventral (orange) stress fibers (*Figure 2A*).

## Plasmids and siRNA Oligonucleotides

DNA transfections were performed as described in *Tojkander et al. (2011)*. The following constructs were used in experiments: wild-type GFP-VASP, GFP-VASP$_{ser239ala,thr278ala}$, which was generated from the triple mutant AAA-GFP-VASP construct (*Benz et al., 2009*), GFP-CaP3 (*Burgstaller et al., 2002*), PA-GFP-actin (a kind gift from Maria Vartiainen), cofilin-1-Flag (*Hotulainen et al., 2005*), Rif-TN, YFP-Tm4, CFP-and YFP-α-actinin, CFP-and mCherry-Zyxin (*Hotulainen and Lappalainen, 2006*), GFP-and Cherry-actin (*Tojkander et al., 2011*), dominant active RhoA (*Vartiainen et al., 2000*). For depletion of VASP, Dharmacon ON-TARGETplus Smartpool cat# L-019763-01, Lot# 121105 was used. For depletion of cofilin-1 target sequence "AAG GAG GAT CTG GTG TTT ATC" was used for a 5′-Alexa Fluor 488 labelled siRNA, which was purchased from Qiagen.

## Immunofluorescece microscopy

Cells were fixed with 4% PFA, washed 3 x with 0.2% Dulbecco/BSA and permeabilized with 0.1% Triton X-100 in TBS. Immunofluorescence stainings were performed as in (*Tojkander et al., 2011*). Images were acquired with a charge-coupled device camera (AxioCam HRm; Zeiss) on a microscope (Axio Imager.M2; Zeiss). AxioVision Rel. 4.8 (Zeiss) and PlanApo 63x/1.40 (oil) objective (Zeiss) was used for the image acquirement. The following reagents and antibodies were used for the stainings: Alexa phalloidin 488, 568, 594 and 647 (1:200–400 dilutions) (Life Technologies™), anti-cofilin-1 antibody (Abcam, ab11062), anti-VASP antibodies (1:50–100) (Sigma, HPA005724 and Enzo, IE273), VASP-phospho-T278 antibody (1:50) (ImmunoWay), VASP-phospho-S239 antibody (1:50) (Millipore, 16C2), anti-vinculin antibody (1:50) (Sigma, hVin-1). DAPI and secondary antibodies, which were conjugated to Alexa Fluor 488, Alexa Fluor 568/594, or Cy5 were from Life Technologies.

## Western blotting

Cells were washed with cold PBS, scraped, and lysed in PBS, 1% Triton X-100 (with 0.3 mM PMSF and protease and phosphatase inhibitor cocktail (Pierce). Protein concentrations were measured using Bradford reagent (Sigma-Aldrich). Alternatively, cells were lysed after washes into 4x LSB-DTT buffer for obtaining total cell lysates. Lysates were briefly sonicated prior to boiling. Mixture of 5% milk/BSA was used for blocking. Following antibodies were used for detection with dilutions recommended by the manufacturers: rabbit polyclonal anti-VASP antibodies (Sigma, HPA005724 and Enzo, IE273), VASP-phospho-T278 antibodies (ImmunoWay and ECM Biosciences, VP2781), VASP-phospho-S239 antibodies (Millipore, 16C2; Sigma SAB4504565; Abcam, 16C2), anti-GAPDH (Sigma, G8795). Appropriate HRP-linked secondary antibodies (Promega) and ECL reagent (AmershamTM, GE Healthcare) were applied for chemiluminescence detection of the blots. Quantity One 4.1.1 program (Bio-Rad) was used to quantify the band intensities of blots.

## Photoactivatable (PA) GFP-Actin

Live cell imaging with PA-GFP-actin/Cherry-Zyxin-transfected U2OS cells was performed as above with 3I Marianas imaging system. Three captures were taken before activation of PA-GFP-actin with 405 lasers. Activation was performed at the adhesion sites at the tips of dorsal and ventral stress fibers. 488 and 561 lasers were used to visualize activated protein and focal adhesion marker Zyxin, respectively. Images were captured 5x every 2 ms, after which the signal was recorded every 20 s.

## Fluorescence recovery after photobleaching, FRAP

For measuring vectorial actin polymerization as well as actin dynamics within focal adhesions by fluorescence recovery after photobleaching (FRAP), cells were transfected with GFP-actin construct and incubated for 24 hr. Prior to imaging, the cells were moved to fibronectin-coated (10 μg/ml) glass-bottomed dishes (MatTek Corporation) and 3I Marianas imaging system (3I intelligent Imaging Innovations) with 63x/1.2 water objective (C-Apochromat Corr WD = 0.28 M27) was used. Five pre-bleach images were acquired before bleaching with 100% intensity of 488 (50 mW) for 1 x 1 ms. First post-bleach images were acquired 10x every 500 ms and after that every 10 s. In laser ablation experiments, five pre-ablation images were acquired and bleaching was performed 10 s after ablation. The rate of vectorial actin polymerization at focal adhesions was determined by a blind analysis (performed from randomly ordered samples by a different person to the one that carried out the experiments and prepared the kymographs) by ImagePro Plus 6.0 software. Here, the speed of

stress fiber elongation was quantified by measuring the advancement rate of proximal ends of the photobleached stress fiber regions form the kymographs prepared from the movies.

## Laser ablation

Ablation of single ventral stress fibers was performed with 100% intensity of 405 nm laser (100mW, in 3I Marianas imaging system) using 3 x 200 ms pulses. Five captures were taken before the ablation, after which retraction of the fibers was followed for 10 s before recording the changes in actin dynamics in relaxed fibers. Growth rate of actin filaments from the adhesions of ablated- or non-ablated ventral stress fibers were followed by either FRAP or photoactivatable-GFP-actin with 3I Marianas as explained above. It is important to note that experiments on cells co-expressing mCherry-Actin and GFP-Zyxin demonstrated that focal adhesions at the ends of ventral stress fibers are immobile after ablation (see *Video 7*). In few cases, laser ablation, however, led to retraction of the cell edge and accompanied disruption of focal adhesions. All such cases were discarded from further analysis.

## Image analyses

Image Pro Plus 6.0 program was used for the quantifications of focal adhesion and stress fiber properties. Dorsal stress fiber lengths from fixed ctrl or ML-7-, Y27632-, KT5823- or Compound C-treated as well as Rif-TN-transfected cells were analysed from at least 12 cells and 3 fibers per each cell (exact numbers for each experiment are indicated in the figure legends). Focal adhesion sizes and angles were quantified from frames of live cell imaging captures. Angles were calculated as the change between the major axis of the focal adhesion and the vertical. Intensity of phospho-VASP antibody stainings as well as co-localizations in the adhesion sites were analysed with line profiles. Intensities of phospho-VASP (ser239 and thr 278) were divided with the intensity of total VASP antibody staining and values for ventral stress fiber adhesions were normalized to 1. Distances of periodic spacing of the transverse arc structures were also measured with line profiles in Image Pro.

## Statistical analyses

Differences between groups were compared using the unpaired student t-test assuming unequal variances. All data were reported as mean +/- SEM or SD as indicated in the figure legends.

## Acknowledgements

We thank Anna-Liisa Nyfors for technical assistance and the Light Microscopy Unit of the Institute of Biotechnology for the help and advice in live-cell imaging. Thomas Renne, Mario Gimona, Johan Peränen and Maria Vartiainen are acknowledged for providing valuable reagents, and Pirta Hotulainen, Johan Peränen and Ville Hietakangas for comments on the manuscript. This study was supported by grants from Sigrid Juselius Foundation (to PL) and Academy of Finland (to ST and PL). GG was supported by fellowship from Viikki Graduate program in Biosciences (VGSB).

## Additional information

### Funding

| Funder | Author |
| --- | --- |
| Suomen Akatemia | Sari Tojkander<br>Pekka Lappalainen |
| Sigrid Juséliuksen Säätiö | Pekka Lappalainen |

The funders had no role in study design, data collection and interpretation, or the decision to submit the work for publication.

### Author contributions

ST, Conception and design, Acquisition of data, Analysis and interpretation of data, Drafting or revising the article, Contributed unpublished essential data or reagents; GG, Acquisition of data, Analysis and interpretation of data; AH, RK, Acquisition of data, Analysis and interpretation of data,

Drafting or revising the article; PL, Conception and design, Analysis and interpretation of data, Drafting or revising the article

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
