## [Decision Letter]

Thank you for sending your work entitled "Generation of contractile actomyosin bundles depends on mechanosensitive actin filament assembly and disassembly" for consideration at *eLife*. Your article has been favorably evaluated by Vivek Malhotra (Senior Editor) and three reviewers, one of whom is a member of our Board of Reviewing Editors.

The Reviewing Editor and the other reviewers discussed their comments before we reached this decision, and the Reviewing Editor has assembled the following comments to help you prepare a revised submission.

This manuscript investigates the mechanisms of stress fiber assembly. The authors focus on the mechanosensitive nature of ventral stress fiber assembly and alignment in migrating cells. The authors show that ventral stress fibers result from transverse arc fusion during centripetal flow. Fusion leads to increased contractility, which in turn inhibits vectorial actin polymerization via AMPK mediated VASP phosphorylation at the associated focal adhesions. Finally, the authors show that ADF/cofilin mediated stress fiber disassembly is also mechanosensitive and that strongly contractile fibers are protected from cofilin.

The reviewers found that the paper provides interesting data and interprets them in the framework of a neat scenario describing the maturation of contractile stress fibers and the role of a mechanosensitive VASP phosphorylation in strengthening adhesion complexes as they mature. However, the reviewers found that the conclusions are often not sufficiently supported by the data and additional experiments would be needed to take the observations from a correlative to a mechanistic level.

Specifically, as it is, the paper does not fully demonstrate its main conclusions on mechanosensitive processes involved in stress fiber assembly and homeostasis. We list below experimental directions suggested by the reviewers. While they probably cannot all be performed within the timeframe of a revision, a subset could be sufficient to demonstrate that vectorial growth and vsf assembly are mechanosensitive and that this is mediated, at least in part, by VASP phosphorylation. Demonstrating these two points would provide a biophysical mechanism of interest to the broad readership of *eLife*.

Additional evidence is needed to demonstrate that the events described are actually mechanosensitive. The mechanosensitivity of VASP phosphorylation and vectorial growth is mostly based on comparison of cell behaviors on substrates of different stiffness. This is rather indirect. A key experiment would be to directly show that decreasing of increasing tension affects actin assembly dynamics at vsf. Could the authors more directly perturb the forces exerted on focal adhesions by vsf, e.g. with local relaxation of tension by laser ablation or local increase in tension by pulling on the cell (like in Riveline et al for instance)? Would such perturbations affect vectorial growth, as visualized with photoactivation of actin-GFP? To further support mechanosensitivity, the authors show that myosin inhibition leads to defects in vsf formation and longer dorsal stress fibers. But shouldn't decreasing tension rather result in longer vsf, as vectorial growth would not be inhibited anymore?

Another direction would be to make a more extensive use of traction force measurements to substantiate comments on stress fibers sensing tension. Statements like "thus, we examined whether ADF/cofilins could be responsible for specific disassembly of those dorsal stress fibers that are not under tension in migrating U2OS cells" or "these experiments revealed that dorsal stress fibers, oriented perpendicularly to the contractile arc, sense weaker myosin II-generated tension and disassemble during stress fiber maturation process (Figure 6)" are rather speculative and direct tension measurements would allow to substantiate them.

That VASP phosphorylation is involved in mechanotransduction is a key point in this paper. The evidence that AMP kinase activity can tune mechanosensitivity is interesting, but relies on the effects of one pharmacological inhibitor (AICAR). A Western blot should be added showing effects on VASP Ser239 and Thr278 phosphorylation after AICAR treatment. To control for specificity, experiments need to be done to test whether the phospho-VASP mutants (used in Figure 4) mimic the effects of AICAR.

Other major points:

In the subsection “Actin polymerization in focal adhesions is controlled by phosphorylation of VASP”, the authors state: "VASP phosphorylation at Ser239 and Thr278 is regulated by cAMP- and cGMP dependent protein kinases PKA and PKG as well as by AMP-activated Protein Kinase…". The authors have focused on AMPK. Why? Did they test examine effects of modulating PKA and/or PKG activity re regulation of actin assembly at focal adhesions and effects on force transduction? If not, what is your rationale for assuming these kinases are not involved?

The differences in actin dynamics in dorsal as compared to ventral stress fibers might derive from reduced actin polymerization (regulated by VASP) in adhesions of ventral stress fibers, but could also result from differences in overall organization of actin filaments and their orientation. Can the authors exclude this possibility? Performing photobleaching experiments at focal adhesions rather than in the middle of the fibers could allow answering this question. Also, could the authors clarify where exactly PA-GFP-actin is activated in Figure 2? Is it only in the adhesion region as highlighted by zyxin? In this case, the signal in the vsf did distribute from the adhesion zone, contrary to what is stated in the legend.

Is the total cellular F-actin amount conserved in the different treatments? Could changes in dsf length and the absence/presence of vsf result from a limiting actin pool?

Y27632 also inactivates LIM kinase (Maekawa et al., Science, 1999), which in turn leads to cofilin activation. Is this the case in U2OS cells? And if so, does it mean that this effect is dominated by the effect of Y27632 on myosin activity here?

Treatments leading to longer/shorter dsf seem to also affect overall cell morphology. Is cell size changed? And if so, shouldn't one normalize dsf length to cell or lamella area? Or does the length of dsf control lamella size?

[Editors’ note: this article was rejected after discussions between the reviewers at resubmission, but the authors were invited to resubmit after an appeal against the decision.]

Thank you for choosing to send your work entitled "Generation of contractile actomyosin bundles depends on mechanosensitive actin filament assembly and disassembly" for consideration at *eLife.* Your full submission has been evaluated by Vivek Malhotra (Senior Editor), a Reviewing Editor and two reviewers, and the decision was reached after discussions between the reviewers. Based on our discussions and the individual reviews below, we regret to inform you that your work will not be considered further for publication in *eLife*.

While the reviewers appreciate that the importance of mechanosensitivity in stress fiber assembly and dynamics, they are concerned that your conclusions are not really supported by the data. The novel conclusion of the paper is a mechanosensitive VASP phosphorylation that regulates vectorial growth of stress fibers. However, as already pointed out in the first round of review, the authors have not really provided definitive data to support their conclusions. The new ablation experiment provides some support (if statistics and clarification on the quantifications are provided, see comments in reviews below), however it does not by itself support a mechanosensitive mechanism. After extensive discussion, the reviewers concluded that the authors should quantify forces if they want to claim mechanosensitivity, and quantitatively assess the relation between forces exerted and actin assembly rates. Furthermore, it is unclear if resumed vectorial polymerization upon ablation actually depends on VASP. This needs to be addressed to rule out other possible explanations. Clarification of these points would be essential to support the claims and it is unclear whether the issues can be addressed in a reasonable time.

Reviewer #2:

The ablation photo-activation/FRAP experiments presented in Figure 3 provide new evidence supporting the author's contention that force reduction promotes vectoral actin assembly. With that said, it would have been much more compelling if the authors had correlated actin assembly and experimental changes in VASP activity with actual forces. Since the authors are using traction force microscopy (Figure 2) it is not clear why this technique was not employed to address this key question.

Figure 6 is improved by the quantitative assessment of phospho-Vasp levels before and after AICAR treatment; however, representative Western blots from which this data was obtained need to be shown.

Reviewer #3:

Having carefully compared the original and revised versions of the manuscript, it seems that only very minor changes have been introduced into the revised version. Indeed, the biggest change is the addition of one experiment that the authors propose to confirm their previous conclusions.

The experiment (shown in Figure 3) is admittedly fancy, as the authors propose that they can introduce the feature of "vectorial actin polymerization", which they propose normally to be specific to dorsal stress fibers, in ventral stress fibers simply by relaxing them through laser-ablation. This is not uninteresting, but as far as I can see, the effect is comparably modest, and the authors show two individual examples only, without providing any statistics on how general or reproducible these two preliminary observations might be.

Notwithstanding this, my biggest problem with understanding the authors' conclusions is as follows: as far as I could follow the model displayed in Figure 9 (Figure 8 in previous version), the authors propose that VASP phosphorylation in adhesions anchoring vsf inhibits what they call "vectorial actin polymerization".

I don't want to repeat what I outlined in my previous review, but I would like to emphasize that I previously raised the concern that the absence of vectorial actin polymerization could be simply caused by the difference in overall organization of actin filaments and their orientation in the two-stress fiber types. Now the authors state in their rebuttal letter that they don't want to examine actin assembly rates in adhesions because they "contain many different actin filament populations", which makes it "difficult to follow vectorial actin polymerization if the photobleaching is performed at the actual adhesion". More importantly, the authors state that they did not find any differences in actin turnover (by GFP-actin recovery) when bleaching adhesions located at the ends of dorsal vs ventral stress fibers. This confirms what I had proposed/feared in my review, which is that actin polymerization rates in adhesions anchoring dorsal vs ventral stress fibers are not significantly different. This also confirms that the modest changes in VASP phosphorylation between the two adhesion types cannot introduce a significant change in actin assembly rates.

Instead, these observations suggest to me that it is the overall organization of actin filaments and their polarity in the two-stress fiber types which causes the differences observed in vectorial actin polymerization. That this might possibly be influenced by releasing the tension due to laser ablation could be interesting, but the data are too preliminary in my view to be published in *eLife*.

But even if it were true, my biggest problem with all these conclusions and the way the model is drawn at present is that it represents a sort of circular argument. Dorsal stress fibers undergo vectorial actin polymerization, which is proposed to be VASP-dependent, whereas ventral stress fibers prominently accumulate VASP, but due to its inhibition by phosphorylation, vectorial actin polymerization is inhibited. Do the authors really want to propose that VASP function is restricted to dorsal stress fibers? Are the authors sure they want to imply that VASP accumulating in adhesions anchoring ventral stress fibers is not functional simply because it is phosphorylated?

Furthermore, if I took the model seriously, I would ask myself how vectorial actin polymerization can be inhibited in focal adhesions anchoring ventral stress fibers if actin assembly is still taking place (as stated in the rebuttal letter), or the other way around, how can active VASP in adhesions anchoring dorsal stress fibers drive vectorial actin polymerization in spite of the presence of additional actin assembly factors potentially present in adhesions (as mentioned by the authors) and along stress fibers? I feel that this model is too simplified to explain the observations described. In my view, essential regulatory components beyond VASP localization and regulation are missing here, but would certainly be required for a comprehensive view of why dorsal and ventral stress fibers display the discussed differences in vectorial actin polymerization.

[Editors’ note: what now follows is the decision letter after the authors submitted for further consideration.]

Thank you for choosing to send your work entitled "Generation of contractile actomyosin bundles depends on mechanosensitive actin filament assembly and disassembly" for consideration at *eLife*. Your letter of appeal has been considered by Vivek Malhotra (Senior Editor) and a Reviewing Editor.

As also mentioned in the appeal letter, the main point to be addressed in the revision was "to demonstrate that vectorial growth and vsf assembly are mechanosensitive and that this is mediated, at least in part, by VASP phosphorylation". While the laser ablation experiments alleviate to some extent the first part of the concern (provided concerns about quantifications can be addressed, as detailed below), the second part remains unclear. The Western blot of VASP phosphorylation after AICAR treatment, which was suggested in another part of the reviewers' comments, is indeed important (though the original Western should be shown in addition to the quantification). However it does not directly link VASP phosphorylation or activity, to tension. This could be addressed by e.g. testing if vectorial growth after laser ablation depends on VASP. Without linking the resumed actin polymerization to VASP activity it seems a stretch to conclude that the release of tension by ablation triggers actin polymerization via VASP.

Concerning the laser ablation experiments, the concern raised after the revision is that it is not clear how quantifications were performed. The actin signal is very dim on the image displayed, was the speed of growth quantified by hand or in an unbiased (automated) manner? If one were to draw a line along the stronger/less patchy actin signal, the speed of growth would be much lower than suggested by the dotted line in the figure. Given the low signal in the figure, providing a movie might also help assess the resumed actin polymerization. Furthermore, the text indicates a mean speed of resumed assembly, but the number of cells or experiments does not seem to be reported. While indeed the option of more extensively using traction forces was only suggested in the initial round of review, given the important of the mechanosensitivity statement, it would considerably strengthen the conclusions of the laser ablation experiment if tractions force microscopy was used to show that it indeed releases tension in the stress fiber.

The appeal letter lists a number of experiments on VASP. However, most of the treatments listed lead to effects on dorsal stress fibers. Yet, one key point of the model proposed is that mechanosensitive VASP phosphorylation stabilizes ventral stress fibers by preventing vectorial growth, as detailed in Figure 9. This is at this point a rather unsupported statement, which the authors might be able address by exploring the role of VASP in the laser ablation experiments, as suggested in the appeal letter.

Concerning the concerns of Reviewer 3, the responses provided in the appeal letter suggest that they could indeed be addressed by further clarifications in the text and a more extensive investigation of the laser ablation experiments. However, this reviewer's concerns about actin turnover at focal adhesions versus vectorial growth are relevant and the distinction may be confusing for many readers as well. It would be particularly important to further clarify this distinction in the text and possibly include the FRAP experiments at the different types of adhesion to make clear that the paper does not mean to claim that VASP phosphorylation stops all actin assembly in vsf (as may be wrongly assumed from a quick look at Figure 9). Again, a thorough investigation of the laser ablation experiments, including the role of VASP and if possible traction force measurements of the force release, could help addressing these concerns.

If these experiments can be added and the points listed above can be addressed, we are prepared to consider a revised submission with no guarantees of acceptance.

[Editors' note: further revisions were requested prior to acceptance, as described below.]

Thank you for resubmitting your work entitled "Generation of contractile actomyosin bundles depends on mechanosensitive actin filament assembly and disassembly" for further consideration at *eLife.* Your revised article has been favorably evaluated by Vivek Malhotra (Senior Editor), Ewa Paluch (Reviewing Editor), and has been discussed with one reviewer. The manuscript has been very much improved and the more thorough investigation of the laser ablation experiments makes the conclusions of the paper much clearer and better supported. However, some minor remaining issues need to be addressed before acceptance, as outlined below:

The traction force microscopy experiment showing that laser ablation actually releases tension in ventral stress fibers (Figure 3—figure supplement 1) is essential, as it demonstrates that the method works. The authors state it is representative, but of how many experiments? Could some quantification of the released forces and of the number of experiments performed be provided?

Do focal adhesions at the end of the stress fibers move upon laser ablation or do they remain immobile? This is important because adhesion movement when vectorial growth is being measured would affect the measured elongation rates. Or is the bleaching performed after potential adhesion movements have relaxed?

Does the quantification of vectorial actin polymerisation rates provided in Figure 7 also correspond to the experiments displayed in Figure 3? If so, please clarify in the figure legend/text. If not, could a similar quantification be provided for Figure 3?

In his last comment, reviewer 3 was rather asking how VASP can promote vectorial growth specifically, i.e. how does VASP elongate only or preferentially filaments pointing towards the cell center. Could the authors speculate on how they envisage this could be achieved at the microscopic level? This is an important point that should be clearly stated in the Discussion of the paper.

In Figure 7, has the kymograph been turned upside down compared to what is displayed in the picture of the whole cell? Otherwise it seems that the resumed growth occurs distally, towards the outside of the cell from the adhesion point at the end of the stress fiber, which does not seem to make sense.

Is there a condition missing in Figure 5 on the left of WT VASP? The space between the y axis and WT VASP is rather wide.

---

## [Author Response]

*Specifically, as it is, the paper does not fully demonstrate its main conclusions on mechanosensitive processes involved in stress fiber assembly and homeostasis. We list below experimental directions suggested by the reviewers. While they probably cannot all be performed within the timeframe of a revision, a subset could be sufficient to demonstrate that vectorial growth and vsf assembly are mechanosensitive and that this is mediated, at least in part, by VASP phosphorylation. Demonstrating these two points would provide a biophysical mechanism of interest to the broad readership of* eLife. *Additional evidence is needed to demonstrate that the events described are actually mechanosensitive. The mechanosensitivity of VASP phosphorylation and vectorial growth is mostly based on comparison of cell behaviors on substrates of different stiffness. This is rather indirect. A key experiment would be to directly show that decreasing of increasing tension affects actin assembly dynamics at vsf. Could the authors more directly perturb the forces exerted on focal adhesions by vsf, e.g. with local relaxation of tension by laser ablation or local increase in tension by pulling on the cell (like in Riveline et al for instance)? Would such perturbations affect vectorial growth, as visualized with photoactivation of actin-GFP? To further support mechanosensitivity, the authors show that myosin inhibition leads to defects in vsf formation and longer dorsal stress fibers. But shouldn't decreasing tension rather result in longer vsf, as vectorial growth would not be inhibited anymore?*

We thank the reviewers for these excellent suggestions, and have therefore performed laser ablation experiments to induce local relaxation of tension in contractile ventral stress fibers. These laser ablation experiments were combined with FRAP and photoactivation assays to follow actin dynamics and ‘vectorial actin polymerization’ in focal adhesions located at the ends of ablated ventral stress fibers and non-ablated ventral fibers within the same cells. The new data are presented in Figure 3 and Figure 3—figure supplement 1, and discussed in the subsection “Tension provided by myosin II inhibits vectorial actin polymerization at focal adhesions”. These experiments revealed that local relaxation of the contractility rapidly induces vectorial actin polymerization at adhesions located at the tips of the stress fiber. Thus, these data provide direct evidence that vectorial actin polymerization in focal adhesions is mechanosensitive (as already suggested by indirect experiments with pharmacological inhibitors in the original version of the manuscript). It should be noted that the rate of stress fiber elongation from focal adhesions varied quite significantly between individual ablation experiments, and this is most likely due to differences in the fiber thickness, state of relaxation, and connections to other contractile stress fibers. However, on average the elongation rates of laser-ablated ventral stress fibers were 0,298 um/min (SD +/- 0,11), which is close to the elongation rate of non-contractile dorsal stress fibers [0,23 um/min (SD +/- 0,048)].

Concerning the experiments with MLCK/ROCK inhibitors, our data presented in Figure 2—figure supplement 2 showed that inhibition of myosin II activity results in depletion of contractile ventral stress fibers and transverse arcs, and an increase in the length of dorsal stress fibers (this is described in “Tension provided by myosin II inhibits vectorial actin polymerization at focal adhesions”). This is consistent with the model where decreased tension results in increased vectoral actin polymerization at focal adhesions and consequent formation of longer dorsal stress fibers.

*Another direction would be to make a more extensive use of traction force measurements to substantiate comments on stress fibers sensing tension. Statements like "thus, we examined whether ADF/cofilins could be responsible for specific disassembly of those dorsal stress fibers that are not under tension in migrating U2OS cells" or "these experiments revealed that dorsal stress fibers, oriented perpendicularly to the contractile arc, sense weaker myosin II-generated tension and disassemble during stress fiber maturation process (Figure 6)" are rather speculative and direct tension measurements would allow to substantiate them.*

Because the laser-ablation experiments described above (and shown in Figure 3 of the revised manuscript) provided direct evidence for the mechanosensitivity of vectorial actin polymerization in focal adhesions, we feel that additional traction force measurements are perhaps no longer necessary to elaborate this conclusion. However, we would like to point out that tension measurements by traction force microscopy presented in our manuscript (Figure 2), and in a new publication by Soine et al., (2015) provide evidence that ventral stress fibers are typically under higher tension compared to non-contractile dorsal stress fibers. This is now discussed in the subheading “Transverse arc fusion is accompanied by increased contractility of actomyosin bundles and alignment of distal focal adhesions”.

*That VASP phosphorylation is involved in mechanotransduction is a key point in this paper. The evidence that AMP kinase activity can tune mechanosensitivity is interesting, but relies on the effects of one pharmacological inhibitor (AICAR). A Western blot should be added showing effects on VASP Ser239 and Thr278 phosphorylation after AICAR treatment. To control for specificity, experiments need to be done to test whether the phospho-VASP mutants (used in Figure 4) mimic the effects of AICAR.*

We have performed a Western blot to confirm that AICAR, a pharmacological activator of AMPK, increases phosphorylation of Ser239 residue on VASP. Quantification of the results from the Western blot analysis is presented in Figure 6. These data, together with experiments performed with pharmacological inhibitors of AMPK and mutant versions of VASP (Figure 4—figure supplement 1 and Figure 5), suggest that AICAR induces maturation of stress fibers through inhibition of actin polymerization activity of VASP.

We also agree that, in principle, it would be interesting to test whether the phosphomimetic VASP mutants would recapitulate the phenotype induced by AICAR. However, because phosphorylation of VASP inhibit its actin polymerization activity (e.g. Harbeck et al., 2000; Benz et al., 2009) the phenotypes of RNAi-rescue cells expressing phosphomimetic VASP mutants instead of wild-type VASP are expected to resemble the phenotype in VASP depleted cells. Indeed, as shown in Figure 4—figure supplement 1 of our manuscript, VASP depletion results in comparable phenotype (lack of dorsal stress fibers) as induced by AICAR treatment. This is now discussed in the subsection “Actin polymerization in focal adhesions is controlled by phosphorylation of VASP”.

Other major points:In the subsection “Actin polymerization in focal adhesions is controlled by phosphorylation of VASP”, the authors state: "VASP phosphorylation at Ser239 and Thr278 is regulated by cAMP- and cGMP dependent protein kinases PKA and PKG as well as by AMP-activated Protein Kinase…". The authors have focused on AMPK. Why? Did they test examine effects of modulating PKA and/or PKG activity re regulation of actin assembly at focal adhesions and effects on force transduction? If not, what is your rationale for assuming these kinases are not involved?

We decided to focus on AMPK-mediated phosphorylation of VASP because earlier studies on muscle cells provided evidence for mechanosensitive activation of AMPK (Blair et al., 2009). However, we have now also performed experiments with available PKA and PKG inhibitors (KT5720 and DT-2, respectively) to study their effects on dorsal stress fiber elongation. These experiments revealed that in KT5720 and DT-2 treated U2OS cells dorsal stress fibers were 1,25 and 1,35 fold longer as compared to control cells that were treated with DMSO alone. Thus, also PKA and PKG may contribute to regulation of VASP activity in U2OS cells, but based on these inhibitor studies their contribution to this process is less pronounced compared to AMPK (cells treated with AMPK inhibitor, compound c, had approximately 1,9 times longer dorsal stress fibers compared to control cells). These new data are discussed in the subheading “Actin polymerization in focal adhesions is controlled by phosphorylation of VASP”. It is also important to note, that AMPK activator AICAR resulted in opposite effects on dorsal stress fibers as compared to AMPK inhibition (or expression of phosphorylation-deficient VASP mutants), providing further support for the central role of AMPK in this process.

*The differences in actin dynamics in dorsal as compared to ventral stress fibers might derive from reduced actin polymerization (regulated by VASP) in adhesions of ventral stress fibers, but could also result from differences in overall organization of actin filaments and their orientation. Can the authors exclude this possibility? Performing photobleaching experiments at focal adhesions rather than in the middle of the fibers could allow answering this question. Also, could the authors clarify where exactly PA-GFP-actin if activated in Figure 2? Is it only in the adhesion region as highlighted by zyxin? In this case, the signal in the vsf did distribute from the adhesion zone, contrary to what is stated in the legend.*

Focal adhesions are expected to contain many different actin filament populations (see e.g. Tojkander et al., 2011), which make it difficult to follow the vectorial actin polymerization if the photobleaching is performed at the actual adhesion. Thus, studies by others (e.g. Tee et al., 2015) and us (e.g. Hotulainen and Lappalainen, 2006) have shown that elongation of dorsal stress fibers through actin polymerization at focal adhesions can be best studied by performing the photobleaching right below the adhesion. This is further clarified in the subheading “Tension provided by myosin II inhibits vectorial actin polymerization at focal adhesions”. However, as requested, we have now also performed FRAP at focal adhesions of GFP-actin expressing cells to demonstrate rapid, relatively uniform recovery of GFP-actin in focal adhesions. We did not observe significant differences in the GFP-actin recovery in adhesions located at the ends of dorsal vs. ventral stress fibers, and thus an example of FRAP experiment on actin dynamics at the dorsal stress fiber linked adhesion is presented in Video 1.

We have now also clarified in the legend to Figure 2 that PA-GFP-Actin was activated at the focal adhesion and just below it. Yellow dashed lines show the borders of the activated region. In the case of ventral stress fiber, the PA-GFP-actin signal does not ‘spread’ below this border, whereas in the case of dorsal stress fiber the PA-GFP-actin signal moves towards the cell center along the elongating dorsal stress fiber.

*Is the total cellular F-actin amount conserved in the different treatments? Could changes in dsf length and the absence/presence of vsf result from a limiting actin pool?*

Because our new laser ablation studies provided direct evidence that vectorial actin filament polymerization in focal adhesion is indeed mechanosensitive (here we compared vectorial actin polymerization in ablated vs. non-ablated ventral stress fiber within the same cell), we feel that trying to quantify the G:F-actin ratios in cells treated with different pharmacological inhibitors is no longer necessary for confirming the main conclusions of the study.

*Y27632 also inactivates LIM kinase (Maekawa et al., Science, 1999), which in turn leads to cofilin activation. Is this the case in U2OS cells? And if so, does it mean that this effect is dominated by the effect of Y27632 on myosin activity here?*

We assume that the effects seen by ROCK inhibitor Y27632 are mainly due to its effects on myosin activity, because we see a decrease in myosin-containing contractile structures. We have not tested the impact of Y27632 on LIM kinase activity and phosphorylation of cofilin-1. However, assuming that Y27632 would decrease cofilin phosphorylation and thus increase its actin filament severing activity, we should rather see decrease in dorsal stress fiber length, not increase as we see now. Furthermore, as described above, new laser ablation experiments provided direct evidence on the mechanosensitivity of actin filament assembly, and thus these indirect ROCK inhibitor studies just provide additional support to this conclusion.

*Treatments leading to longer/shorter dsf seem to also affect overall cell morphology. Is cell size changed? And if so, shouldn't one normalize dsf length to cell or lamella area? Or does the length of dsf control lamella size?*

We have now examined the effects of different pharmacological treatments on cell size, and learned that they result in 1.0 – 1,78 fold increase in the cell area. This is now mentioned in the Results. However, it is important to note that contractile stress fibers are important regulators of cell morphogenesis, and thus increase in the cell-size is expected upon depletion of these actomyosin bundles. Furthermore, we sincerely feel that increase in total cell area does not affect the interpretation of these data, because we are not aware of any studies linking total cell area to stress fiber elongation. In contrast, a recent study by Burnette et al. (J Cell Biol., 2014) demonstrated that depletion of contractile transverse arcs decreases rather than increases the lamella width.

[Editors’ note: the author responses to the second round of peer review follow.]

We appreciate that the reviewers have extensively discussed our revisions, but nevertheless find the decision both surprising and unjustified. We would fully understand the reviewers’ comments and the rejection if we would not have performed the suggested experiments, if the assays/analyses would not have been properly carried out, or if the data would not have supported the conclusions presented. However, none of these is the case here.

In the original decision letter we were advised to provide additional support for the mechanosensitivity of vectorial actin polymerization, and the role of VASP phosphorylation in the process: ‘While they probably cannot all be performed within the timeframe of a revision, a subset could be sufficient to demonstrate that vectorial growth and VSF assembly are mechanosensitive and that this is mediated, at least in part, by VASP phorphorylation. Demonstrating these two points would provide a biophysical mechanism of interest to the broad readership of *eLife*.’

Concerning the first point, we were advised to perform either laser ablation or cell pulling experiments, or make more extensive use of traction force measurements. From the three options, we chose to perform laser ablation experiments, which provided direct evidence that vectorial actin polymerization in focal adhesion is indeed mechanosensitive. Unlike stated in the decision letter of the revised manuscript, these assays were rigorously performed; vectorial actin polymerization rates were monitored both by photoactivation and photobleaching assays, and the rates were quantified from several individual ablation experiments. As described in the manuscript, the average the elongation rates of laser-ablated ventral stress fibers were 0,298 um/min (SD +/- 0,11). The statistical significance for the differences in the elongation rates of intact ventral stress fibers and laser ablated ones is (p <0.001; t-test). This can be added either to the text or the quantification can be included as a new bar graph.

Concerning the second point, the reviewers advised us to add a Western blot showing effects on VASP Ser239 and Thr278 phosphorylation after AICAR treatment. To control for specificity, we were also asked to perform experiments to test whether the phospho-VASP mutants (used in Figure 4) mimic the effects of AICAR. However, because both of these experiments were already included in the original version of the text, we sincerely could not figure out what additional assays/analyses could still be performed to further address this point.

We find unclear what additional experiments we could still perform to address these two issues: ‘The reviewers concluded that the authors should quantify forces if they want to claim mechanosensitivity, and quantitatively assess the relation between forces exerted and actin assembly rates. Furthermore, it is unclear if resumed vectorial polymerization upon ablation actually depends on VASP’. Concerning the mechanosensitivity of vectorial actin polymerization, our data (traction force, FRAP and photoactivation experiments) already demonstrated that adhesions at the ends of dorsal stress fibers, which undergo vectorial polymerization, are under lower tension compared to the ones at the ends of ventral stress fibers, which do not undergo vectorial actin polymerization. Furthermore, our laser ablation experiments demonstrated that releasing the tension by disrupting the ventral stress fiber restores vectorial actin polymerization at the tips of the stress fiber. The only additional experiment that we can imagine to be informative would be to measure the tension at the tips of ablated vs. non-ablated ventral stress fibers. If requested, we will be glad to perform such experiment.

Regarding the role of VASP, our experiments already demonstrated that: 1) Levels of phosphorylated VASP correlate with vectorial actin polymerization in focal adhesions, 2) VASP depletion leads to loss of dorsal stress fibers, 3) Expression of phosphorylation-deficient VASP leads to uncontrolled elongation of dorsal stress fibers, 4) Elevated VASP phosphorylation through AMPKactivation leads to loss of dorsal stress fibers, and 5) Diminished VASP phosphorylation through AMPK-inhibition leads to uncontrolled elongation of dorsal stress fibers. Therefore, it is again difficult for us to comprehend, which additional experiments would still be required to demonstrate that mechanosensitive actin filament assembly at focal adhesions is at least partially mediated by VASP phosphorylation (as discussed in the text, we agree with the reviews that in addition to VASP phosphorylation, other mechanosensitive pathways are like to contribute to this process). Here, we could perhaps still preform experiments with different combinations of VASP mutants and AMPK activators/inhibitors, and/or try to perform laser ablation experiments on cells where VASP is inhibited by AICAR treatment to further clarify the role of VASP in this process.

Finally, reviewer #3 listed a number of points, which we do not agree with and/or find unjustified. Thus, I have included our responses to these points in the end of the letter.

For the reasons described above, we would very much appreciate if the Reviewing Editor and the Senior Editor handling this manuscript could carefully re-consider the suitability of our revised manuscript for publication in *eLife*. As explained above, we will be glad to substantially modify the manuscript text as well as perform possible additional experiments if these are considered to be important for providing additional support for the conclusions presented.

Reviewer #2: *The ablation photo-activation/FRAP experiments presented in Figure 3 provide new evidence supporting the author's contention that force reduction promotes vectoral actin assembly. With that said, it would have been much more compelling if the authors had correlated actin assembly and experimental changes in VASP activity with actual forces. Since the authors are using traction force microscopy (Figure 2) it is not clear why this technique was not employed to address this key question.*

As described above, we were originally requested to use either laser ablation, cell pulling or traction force microscopy to address this point. However, if necessary we will be glad to perform new traction force experiments to measure the force at the tips of ablated vs. non-ablated ventral stress fibers as well as try to carry out laser ablation experiments (combined with FRAP) under conditions where VASP is constitutively phosphorylated.*Figure 6 is improved by the quantitative assessment of phospho-Vasp levels before and after AICAR treatment; however, representative Western blots from which this data was obtained need to be shown.*

This can be easily added in the figure.Reviewer #3: *Having carefully compared the original and revised versions of the manuscript, it seems that only very minor changes have been introduced into the revised version. Indeed, the biggest change is the addition of one experiment that the authors propose to confirm their previous conclusions.*

*The experiment (shown in Figure 3) is admittedly fancy, as the authors propose that they can introduce the feature of "vectorial actin polymerization", which they propose normally to be specific to dorsal stress fibers, in ventral stress fibers simply by relaxing them through laser-ablation. This is not uninteresting, but as far as I can see, the effect is comparably modest, and the authors show two individual examples only, without providing any statistics on how general or reproducible these two preliminary observations might be.*

We disagree that the effect is modest, because the rate of vectorial elongation of ablated ventral stress fibers was 0,298 (+/- 0,11) um/min, whereas non-ablated ventral stress fibers displayed elongation rate of 0,014 (+/-0,016) um/min. The average elongation rate in ablated ventral stress fibers was thus comparable to dorsal stress fiber elongation rate (0, 23 (+/- 0,048) um/min). We also do not show individual examples only, but, as described in paragraph 3 of the subheading “Tension provided by myosin II inhibits vectorial actin polymerization at focal adhesions”, we measured the average rate of elongation and provide the SD value for the data. The statistical significance of the difference between the elongation rate of ablated vs. non-ablated ventral stress fibers is (p <0.001; t-test) and this can be added to the text or the data can be presented as an additional bar graph.

*Notwithstanding this, my biggest problem with understanding the authors' conclusions is as follows: as far as I could follow the model displayed in Figure 9 (Figure 8 in previous version), the authors propose that VASP phosphorylation in adhesions anchoring vsf inhibits what they call "vectorial actin polymerization". I don't want to repeat what I outlined in my previous review, but I would like to emphasize that I previously raised the concern that the absence of vectorial actin polymerization could be simply caused by the difference in overall organization of actin filaments and their orientation in the two-stress fiber types. Now the authors state in their rebuttal letter that they don't want to examine actin assembly rates in adhesions because they "contain many different actin filament populations", which makes it "difficult to follow vectorial actin polymerization if the photobleaching is performed at the actual adhesion".*

As already described in the manuscript, several lines of evidence suggest that focal adhesions harbor at least three different actin filament populations with distinct dynamic properties. 1) Three distinct tropomyosins (each one of which likely decorates different actin filaments) are present in focal adhesions of U2OS cells. Tm1 and Tm5NM exclusively reside in focal adhesions, whereas Tm2 decorates the entire dorsal stress fiber and is thus most likely associated with the actin filament population undergoing vectorial actin polymerization (Tojkander et al.,2011). 2) Several proteins involved in actin filament polymerization (e.g. VASP, INF2, Dia1) have been implicated in regulating actin dynamics in focal adhesions. 3) FRAP experiments performed on focal adhesions show rapid uniform GFP-actin recovery (which most likely corresponds to the focal adhesion resident actin filament populations), whereas FRAP experiments performed below focal adhesion (in the region that does not contain TM1 and Tm5MN decorated actin filaments) show treadmilling-like recovery that is indicative of vectorial actin polymerization. We will be glad to further clarify this in the manuscript.

More importantly, the authors state that they did not find any differences in actin turnover (by GFP-actin recovery) when bleaching adhesions located at the ends of dorsal vs ventral stress fibers. This confirms what I had proposed/feared in my review, which is that actin polymerization rates in adhesions anchoring dorsal vs ventral stress fibers are not significantly different.

These data show that rapidly turning over of actin population(s) within focal adhesions display similar dynamics in dorsal vs. ventral stress fibers. However, as very clearly shown by others and us through photoactivation and FRAP experiments performed below adhesions, the actin filament population responsible for vectorial polymerization displays dramatically different behaviors in dorsal vs. ventral stress fibers.

This also confirms that the modest changes in VASP phosphorylation between the two adhesion types cannot introduce a significant change in actin assembly rates.

Based on our data VASP has an important role in vectorial actin polymerization, and its regulation by phosphorylation is critical for the process (as demonstrated by RNAi, mutant VASP expression, AMPK activation and AMPK inhibition experiments).

*Instead, these observations suggest to me that it is the overall organization of actin filaments and their polarity in the two-stress fiber types which causes the differences observed in vectorial actin polymerization. That this might possibly be influenced by releasing the tension due to laser ablation could be interesting, but the data are too preliminary in my view to be published in* eLife.

We find it extremely unlikely that laser ablation would rapidly alter the polarity of heavily cross-linked actin filaments within stress fibers close to focal adhesions (far from the ablated region), and we do not have any data that would support such option. Furthermore, we would like to point out that the laser ablation experiments were combined with both photoactivation and FRAP to examine the stress fiber elongation, and the average rates were rigorously quantified from the FRAP data (see the third paragraph of “Tension provided by myosin II inhibits vectorial actin polymerization at focal adhesions”). We will be also glad to include the statistics of the photoablation experiment in the text or as a new bar graph.

*But even if it were true, my biggest problem with all these conclusions and the way the model is drawn at present is that it represents a sort of circular argument. Dorsal stress fibers undergo vectorial actin polymerization, which is proposed to be VASP-dependent, whereas ventral stress fibers prominently accumulate VASP, but due to its inhibition by phosphorylation, vectorial actin polymerization is inhibited. Do the authors really want to propose that VASP function is restricted to dorsal stress fibers? Are the authors sure they want to imply that VASP accumulating in adhesions anchoring ventral stress fibers is not functional simply because it is phosphorylated?*

Based on our data, VASP promotes vectorial actin filament assembly at the distal tips of dorsal stress fibers, but its actin polymerization activity is inhibited at the adhesions located at the tips of ventral stress fibers. It is important to note that VASP may display other biochemical activities, which may not be affected by phosphorylation (e.g. actin filament bundling) in adhesions located at the ends of ventral adhesions. We agree that this could be further clarified in the manuscript.

*Furthermore, if I took the model seriously, I would ask myself how vectorial actin polymerization can be inhibited in focal adhesions anchoring ventral stress fibers if actin assembly is still taking place (as stated in the rebuttal letter), or the other way around, how can active VASP in adhesions anchoring dorsal stress fibers drive vectorial actin polymerization in spite of the presence of additional actin assembly factors potentially present in adhesions (as mentioned by the authors) and along stress fibers? I feel that this model is too simplified to explain the observations described. In my view, essential regulatory components beyond VASP localization and regulation are missing here, but would certainly be required for a comprehensive view of why dorsal and ventral stress fibers display the discussed differences in vectorial actin polymerization.*

This figure represents a working model for mechanosensitive actin filament assembly and disassembly of stress fibers, and in our opinion provides the most likely explanation for the data presented in our manuscript and earlier publications. We agree with the reviewer that also other pathways (and actin nucleators and polymerization proteins) are likely to contribute to this process and will be glad to clarify this in the figure legend (this was already discussed in the Discussion). Finally, we would like to point out that existing data indicate that focal adhesions are composed of several different actin filament populations and that only the Tm2-decorated one is likely to be involved in vectorial actin polymerization and dorsal stress fiber elongation.

[Editors’ note: what follows is the response to the invitation to resubmit.]

*As also mentioned in the appeal letter, the main point to be addressed in the revision was "to demonstrate that vectorial growth and vsf assembly are mechanosensitive and that this is mediated, at least in part, by VASP phosphorylation". While the laser ablation experiments alleviate to some extent the first part of the concern (provided concerns about quantifications can be addressed, as detailed below), the second part remains unclear. The Western blot of VASP phosphorylation after AICAR treatment, which was suggested in another part of the reviewers' comments, is indeed important (though the original Western should be shown in addition to the quantification). However it does not directly link VASP phosphorylation or activity, to tension. This could be addressed by e.g. testing if vectorial growth after laser ablation depends on VASP. Without linking the resumed actin polymerization to VASP activity it seems a stretch to conclude that the release of tension by ablation triggers actin polymerization via VASP.*

This is an excellent suggestion, and we thus performed ablation experiments also on VASP-depleted cells. The new data revealed that vectorial actin polymerization of ablated ventral stress fibers is diminished by approximately 3-fold in VASP depleted cells compared to control cells (p<0,001). Thus, these data provide direct evidence that VASP is indeed important for mechanosensitive actin filament assembly at focal adhesions. The new data are presented in Figure 7 and discussed in subheading “ADF/cofilin-mediated disassembly of non-contractile actin filament bundles is important for maturation of ventral stress fibers”. Furthermore, a Western blot is now included in Figure 6 as requested.

*Concerning the laser ablation experiments, the concern raised after the revision is that it is not clear how quantifications were performed. The actin signal is very dim on the image displayed, was the speed of growth quantified by hand or in an unbiased (automated) manner? If one were to draw a line along the stronger/less patchy actin signal, the speed of growth would be much lower than suggested by the dotted line in the figure. Given the low signal in the figure, providing a movie might also help assess the resumed actin polymerization. Furthermore, the text indicates a mean speed of resumed assembly, but the number of cells or experiments does not seem to be reported.*

We have now repeated all FRAP experiments concerning vectorial actin filament assembly at focal adhesions with identical microscopy settings, and performed data analysis in an unbiased manner. As described in Material and methods, the speed of elongation was manually quantified from the proximal end of the photobleached region using ImagePro Plus 6.0 software. To avoid any bias, we performed ‘blind analysis’ of the data in a way that elongation speeds were quantified from randomly ordered samples by a different person to the one that performed the experiment and prepared the kymographs. We now also report the number of experiments in the figure legends (see: Figure 2, Figure 3, and Figure 7) and demonstrate that the differences in the vectorial actin polymerization rates between ablated vs. non-ablated and control ablated vs. VASP-depletion ablated stress fibers are highly statistically significant (in all cases p>0,001; t-test).

While indeed the option of more extensively using traction forces was only suggested in the initial round of review, given the important of the mechanosensitivity statement, it would considerably strengthen the conclusions of the laser ablation experiment if tractions force microscopy was used to show that it indeed releases tension in the stress fiber.

We have now performed traction force microscopy combined with laser ablation of ventral stress fibers. A representative example of such experiment, shown in Figure 3—figure supplement 1, demonstrates that laser ablation of a ventral stress fiber indeed releases the tension at focal adhesions located at the tips of ventral stress fibers.

*The appeal letter lists a number of experiments on VASP. However, most of the treatments listed lead to effects on dorsal stress fibers. Yet, one key point of the model proposed is that mechanosensitive VASP phosphorylation stabilizes ventral stress fibers by preventing vectorial growth, as detailed in Figure 9. This is at this point a rather unsupported statement, which the authors might be able address by exploring the role of VASP in the laser ablation experiments, as suggested in the appeal letter.*

As described above, we have now performed laser ablation experiments also on ventral stress fibers of VASP-depleted cells. These experiments revealed that knockdown of VASP results in ~3-fold (p<0,001) decrease in vectorial actin polymerization at focal adhesions of ablated ventral stress fibers as compared to control cells. The new data are shown in Figure 7 and discussed in the subsection “Actin polymerization in focal adhesions is controlled by phosphorylation of VASP”

Concerning the concerns of Reviewer 3, the responses provided in the appeal letter suggest that they could indeed be addressed by further clarifications in the text and a more extensive investigation of the laser ablation experiments. However, this reviewer's concerns about actin turnover at focal adhesions versus vectorial growth are relevant and the distinction may be confusing for many readers as well. It would be particularly important to further clarify this distinction in the text and possibly include the FRAP experiments at the different types of adhesion to make clear that the paper does not mean to claim that VASP phosphorylation stops all actin assembly in vsf (as may be wrongly assumed from a quick look at Figure 9). Again, a thorough investigation of the laser ablation experiments, including the role of VASP and if possible traction force measurements of the force release, could help addressing these concerns.

If these experiments can be added and the points listed above can be addressed, we are prepared to consider a revised submission with no guarantees of acceptance.

As suggested, we have now compared actin dynamics at focal adhesions located at the tips of dorsal vs. ventral stress fibers. These data (Figure 2—figure supplement 2) demonstrate that the turnover of the mobile actin fraction is very similar in dorsal and ventral stress fibers, but there is a small difference in the sizes of immobile fractions. This may be due to absence of vectorial actin actin polymerization in focal adhesions at the tips of ventral stress fibers. However, because the difference was rather modest and there are also alternative explanations for this result, we decided not to elaborate this in the manuscript.

In addition, we have modified manuscript text (please see the subsection of the Resulst “Tension provided by myosin II inhibits vectorial actin polymerization at focal adhesions” and the Discussion) to clarify that: 1) Based on current evidence (including the FRAP analysis described above), focal adhesions are likely composed of several different actin filament populations (specified by different tropomyosins), and 2) VASP phosphorylation does not stop all actin polymerization at focal adhesions, but is specific for the actin filament population responsible for vectorial actin filament assembly (and stress fiber elongation). Finally, we have modified the legend to Figure 10 to specify that focal adhesions are most likely composed of multiple actin filament populations with different dynamics, and that in the figure we only included the actin filament population that is responsible for vectorial actin polymerization (and elongation of stress fibers).

[Editors' note: further revisions were requested prior to acceptance, as described below.]

*The traction force microscopy experiment showing that laser ablation actually releases tension in ventral stress fibers (Figure 3—figure supplement 1) is essential, as it demonstrates that the method works. The authors state it is representative, but of how many experiments? Could some quantification of the released forces and of the number of experiments performed be provided?*

Laser ablation induced tension release was analyzed from three cells, where we ablated one stress fiber and measured the tension around adhesions located at each end of the ablated stress fiber. This analysis demonstrated that ablation resulted in 27,3 +/- 7,0 (SD)% decrease in tension at the region around the adhesion immediately following laser ablation of the fiber. This is now discussed in the legend to Figure 3—figure supplement 1. Please, note that because spatial resolution of the method does not allow measuring the force applied only to the actual adhesion (and thus nearby adhesions or by adjacent non-ablated stress fibers may contribute/compensate traction forces measured), this method most likely underestimates the magnitudes of tension release following stress fiber ablation.

*Do focal adhesions at the end of the stress fibers move upon laser ablation or do they remain immobile? This is important because adhesion movement when vectorial growth is being measured would affect the measured elongation rates. Or is the bleaching performed after potential adhesion movements have relaxed?*

We have performed several laser ablation experiments on cells expressing mCherry-zyxin (in addition to GFP-actin), and these data show that focal adhesions at the ends of ventral stress fibers are immobile also after ablation. In few cases, laser ablation led to retraction of cell edge and accompanied disruption of focal adhesions. All such cases were discarded from the

analysis of vectorial actin growth. This is now explained in the Materials and methods. If necessary, we can also include a new supplementary movie demonstrating that focal adhesions remain immobile after laser ablation.

*Does the quantification of vectorial actin polymerisation rates provided in Figure 7 also correspond to the experiments displayed in Figure 3? If so, please clarify in the figure legend/text. If not, could a similar quantification be provided for Figure 3?*

This is indeed the case, and this is now clarified in the legend to Figure 3.

*In his last comment, reviewer 3 was rather asking how VASP can promote vectorial growth specifically, i.e. how does VASP elongate only or preferentially filaments pointing towards the cell center. Could the authors speculate on how they envisage this could be achieved at the microscopic level? This is an important point that should be clearly stated in the Discussion of the paper.*

In filopodia, VASP promotes the assembly of unipolar actin filament bundles with their barbed ends facing the tip of these protrusions. We believe that VASP similarly catalyzes the assembly of unipolar actin filament bundles in focal adhesions (with rapidly growing barbed ends facing the adhesion, where VASP localizes and where hence the addition to new actin monomers occurs). This is now clarified in the Discussion.

In Figure 7, has the kymograph been turned upside down compared to what is displayed in the picture of the whole cell? Otherwise it seems that the resumed growth occurs distally, towards the outside of the cell from the adhesion point at the end of the stress fiber, which does not seem to make sense.

This is correct. For clarity, we have now rotated the cell image in Figure 7 180^o^ counterclockwise. *Is there a condition missing in Figure 5 on the left of WT VASP? The space between the y axis and WT VASP is rather wide.*

We have now compressed the graph by removing unnecessary space from the left and right hand sides of the bars.